# High sensitivity pressure and temperature quantum sensing in pentacene-doped p-terphenyl single crystals

Harpreet Singh [1,2], Noella D'Souza[1,3], Joseph Garrett[1], Angad Singh [1], Brian Blankenship[1], Emanuel Druga[1], Riccardo Montis [4], Liang Z. Tan [5] & Ashok Ajoy [1,3,6] ✉

Quantum sensors' responsiveness to their physical environment enables detection of variables such as temperature (T), pressure (P), and strain. We present a molecular platform for PT sensing using para-terphenyl crystals doped with pentacene (PDP), leveraging optically detected magnetic resonance (ODMR) of photoexcited triplet electron spins. We observe maximal frequency variations of $df/dP$=1.8 MHz/bar from 0-8 bar and $df/dT$=247 kHz/K from 79–330 K, over 1200 times and threefold greater, respectively, than those seen with nitrogen-vacancy centers in diamond and > 85-fold greater pressure sensitivity over the previous record. Density functional theory calculations indicate picometer-level PT-induced molecular orbital shifts are measurable via ODMR. PDP offers additional advantages including high sensor doping levels, narrow ODMR linewidths, high contrast, and low-cost single crystal growth. Overall, this work reports low-cost, optically-interrogated PT sensors and lays the foundation for increased versatility of quantum sensors through synthetic molecular design.

Quantum sensors are revolutionizing the precise measurement of various physical quantities because of their inherent sensitivity to their environment[1]. While sensors constructed from electronic spins, such as Nitrogen Vacancy (NV) centers in diamond[2,3], are widely used as sensitive magnetic field sensors [4–7], there is growing interest in their ability to probe other parameters, particularly temperature[8,9], pressure[10], strain[11,12], electric field[13], and rotation[14,15].

Sensing with NV centers and other quantum sensors leverages the sensitivity of the triplet-state zero-field splitting (ZFS), $D$, to temperature, pressure or strain, enabling their local, sub-micron-scale, measurement[16,17]. Applications include nanoscale thermometry in single cells[18–20] and probing phase transitions of condensed matter systems in high-pressure anvil cells[21–24]. Optical interrogation of these sensors enables diffraction-limited, non-

invasive sensing – capabilities often lacking in classical sensors (e.g., thermocouples).

Material properties, however, impose an overall bound on achievable sensitivity. For diamond NV centers the slope of variations with temperature and pressure are respectively, $\frac{\partial D}{\partial T}$ = 71 kHz/K and $\frac{\partial D}{\partial P}$ = 1.46 kHz/bar[25,26]; the rigidity of the diamond lattice results in a relatively weak pressure (and strain) sensitivity[11,27]. Downstream implications include an increase in the technical complexity required for manipulating spins via strain[12,28–30]. This motivates exploration of alternative materials that also host a spin-optical interface, similar to NV centers, while offering an enhanced sensitivity to these physical parameters.

Recent advances have highlighted the potential of molecular systems for quantum sensing, utilizing rare-earth or transition-metal

[1]Department of Chemistry, University of California, Berkeley, CA, USA. [2]Department of Physics, Guru Nanak Dev University, Amritsar, Punjab, India. [3]Chemical Sciences Division, Lawrence Berkeley National Laboratory, Berkeley, CA, USA. [4]Dipartimento di Scienze Pure e Applicate (DiSPEA), Università degli Studi di Urbino Carlo Bo, Urbino, Italy. [5]Molecular Foundry, Lawrence Berkeley National Laboratory, Berkeley, CA, USA. [6]CIFAR Azrieli Global Scholars Program, 661 University Ave, Toronto, Canada. ✉e-mail: ashokaj@berkeley.edu

ions[31–34] or photoexcited organic radicals[35,36]. These systems offer advantages stemming from bottom-up synthesis[37], tunable sensor placement in three-dimensions via integration into porous materials[38] and molecular-level control over sensor properties[39].

As a prototypical unit, we recently demonstrated that pentacene molecules doped in para (p)-terphenyl exhibit excellent spin-optical properties and can be exploited for optical magnetometry at room temperature (RT)[40,41]. The photoexcited triplet electron spin can be optically initialized and possesses state-dependent fluorescence contrast, yielding narrow-linewidth optically detected magnetic resonance (ODMR) spectra at RT[41]. Additionally, the material can be grown into large single crystals (3 cm) with high doping levels ( ≈1000 ppm), relative to defects in semiconductor materials, and low concentration of background paramagnetic impurities.

In comparison to defects in semiconductor materials like diamond, the weak, easily deformable, p-terphenyl lattice suggests that this material might exhibit heightened sensitivity to pressure and temperature. In this paper, we show this through a systematic study of photoexcited triplet ODMR spectra across a wide range of temperatures (79–330 K) and pressures (0-8 bar). We measure a pressure and temperature slope >1200-fold and >3-fold greater than that of NV centers respectively, besides other operational advantages. First-principles DFT calculations support experimental findings, provide insight into origins of the enhanced sensitivity, and suggest potential for further improvements in designer chemical systems. The sample is a single-crystal of pentacene doped p-terphenyl (PDP), doped at the 0.1% level. Figure 1A illustrates the lattice structure; $\hat{\mathbf{x}}$ denotes the molecular long-axis, with $\hat{\mathbf{y}}, \hat{\mathbf{z}}$ transverse to it. Crystals were grown using the Bridgman technique[42,43] after zone-refining the p-terphenyl host and subliming pentacene for purification (see Supplementary Note 1). Doping levels exceed those of defects in semiconductor materials (e.g., diamond) by at least two orders of magnitude. The PDP crystals can be grown up to several cm at low cost. Figure 1C compares typical PDP crystal sizes with diamond. The PDP crystal in Fig. 1C requires only $2.06 in materials cost, representing a ~70,000-fold reduction in mass-normalized cost compared to NV-diamond. Polycrystalline material can be obtained by crushing single crystals or inducing imperfect growth to form mm-scale domains.

Figure 1 B shows the energy level diagram of the pentacene π-electron in the p-terphenyl host. It includes a ground state singlet ($|S_0\rangle$), an excited state singlet ($|S_1\rangle$), and a metastable triplet state ($|T_1\rangle$) represented by $|T_x\rangle, |T_y\rangle, |T_z\rangle$, with lifetimes of ~35, 166, and 500 $\mu s$[44]. The photoexcited triplet state is described by the spin Hamiltonian $\mathscr{H}_{sys} = D(S_z^2 - \frac{2}{3}) + E(S_x^2 - S_y^2)$ (1), where $\mathbf{S}$ is a spin-1 Pauli operator, with ZFS parameters $D \approx 1392$ MHz and $E \approx 53$ MHz[45].

Optical excitation populates the triplet state via intersystem crossing (ISC) as $|S_1\rangle \rightarrow |T_{x,y,z}\rangle$ (see Fig. 1B); with pulsed excitation, the $|T_x\rangle$ state is polarized to ≈76%[46]. By selecting appropriate delays, spin state-dependent fluorescence contrast can be obtained exploiting differential relaxation from $|T_1\rangle \rightarrow |S_0\rangle$, in a manner that is conditional on the triplet sub-levels[40,41]. This produces an ODMR spectrum (Fig. 1D, E). Figure 1D is the case for RT and Earth's field. Three transitions are marked, hosting narrow linewidths, here $\ell_0 \approx 4.3$ MHz, even at the high doping level and with power broadening[41]. The spectra here are with illumination with a CW laser; pulsed laser excitation can yield higher ODMR contrast ( ~17%)[41]. Asymmetric lineshapes are influenced by hyperfine couplings to neighboring proton nuclei. Inversion of contrast for the $T_{yz}(|T_y\rangle \leftrightarrow |T_z\rangle)$ transitions is due to higher steady-state population in $|T_z\rangle$ than $|T_y\rangle$[41]. For $T_{xy}$, the electronic coherence time was measured at $T_2^{DD} \approx 18$ $\mu s$ under dynamic decoupling, with $T_1 = 23$ $\mu s$, dominated by triplet-ground relaxation[41].

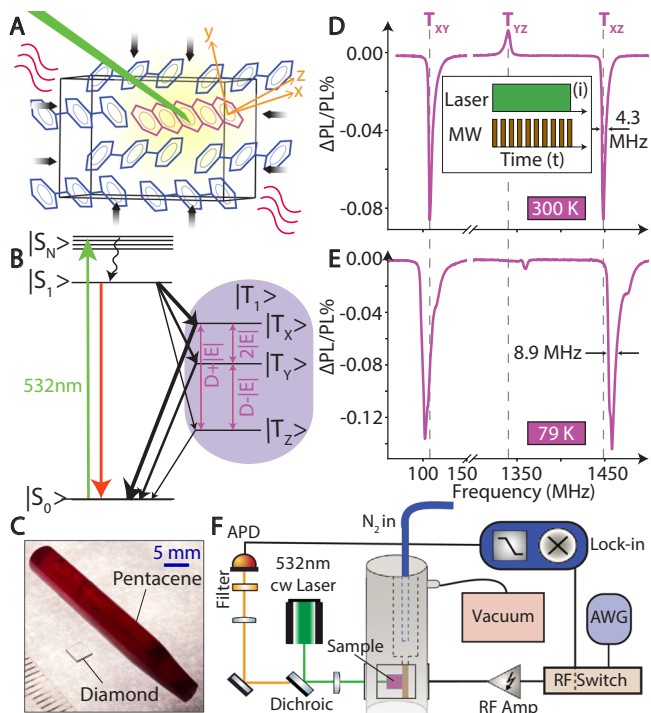

**Fig. 1 | System and principle. A** *System*: Unit cell of optically-interrogated pentacene doped p-terphenyl (PDP). Molecular principal axes are denoted as $\hat{\mathbf{x}}, \hat{\mathbf{y}}, \hat{\mathbf{z}}$. Wavy lines and arrows denote applied changes in temperature and pressure. Laser beam is shown for pentacene photoexcitation. **B** *Energy level diagram* of the pentacene molecule, showing a ground singlet state $|S_0\rangle$, the first excited singlet state $|S_1\rangle$, N higher excited singlets $|S_N\rangle$, and a metastable triplet manifold $|T_1\rangle$---with sublevels $|T_x\rangle, |T_y\rangle$, and $|T_z\rangle$---defined by ZFS parameters $D$ and $E$. Arrow thickness schematically illustrates the differential rates of $|T_1\rangle$ sub-level population and depopulation. **C** *Low-cost cm-scale* PDP crystals are shown. A typical commercial NV-diamond sample is shown alongside for comparison (see scale bar). **D, E** *Representative ODMR spectra* at zero-field and at (**D**) 300 K and (**E**) 79 K, showing marked triplet transitions $T_{xy}$, $T_{yz}$, and $T_{xz}$ with narrow spectral lines. Peaks blue shift and broaden as temperature decreases. Vertical dashed lines serve as a visual guide. Inset (i) shows the initialization pulse sequence employed to obtain all ODMR sensing data. Continuous wave (CW) 532nm laser and 1 kHz modulated microwaves (MW) are both continuously applied to the sample. **F** *Experimental setup* includes a cryostat or pressure chamber housing the sample, a 532 nm CW laser for illumination, and fluorescence collection into an Avalanche photo-diode (APD) via a dichroic mirror. Microwaves are produced by an arbitrary waveform generator (AWG) and delivered via a self-shorted loop.

As schematically shown in Fig. 1A, we investigate changes in the ODMR spectra under varying temperature or pressure, which affect the host lattice and the ZFS parameters, $D$ and $E$. Figure 1F illustrates the experimental setup. The sample is placed in a variable-temperature flow cryostat (Janis ST100) or a pressure chamber. Experiments are conducted on a sub-ensemble of ~$10^9$ pentacene molecules over $2.6 \times 10^{-5}$ mm$^3$.

Figure 1 E shows a typical result. Compared to the RT ODMR spectrum (Fig. 1D), lowering the temperature to 79 K causes a noticeable shift in the positions of the three transitions, as highlighted by the dashed vertical lines.

## Results

### Triplet ODMR variation with temperature

To investigate the spectral changes with temperature, Fig. 2 presents data across a wide range (79-330 K). Figure 2A(i, ii) shows individual ODMR traces for $T_{xy}$ and $T_{xz}$, with color gradients (blue-to-red) representing increasing temperatures. The temperatures are

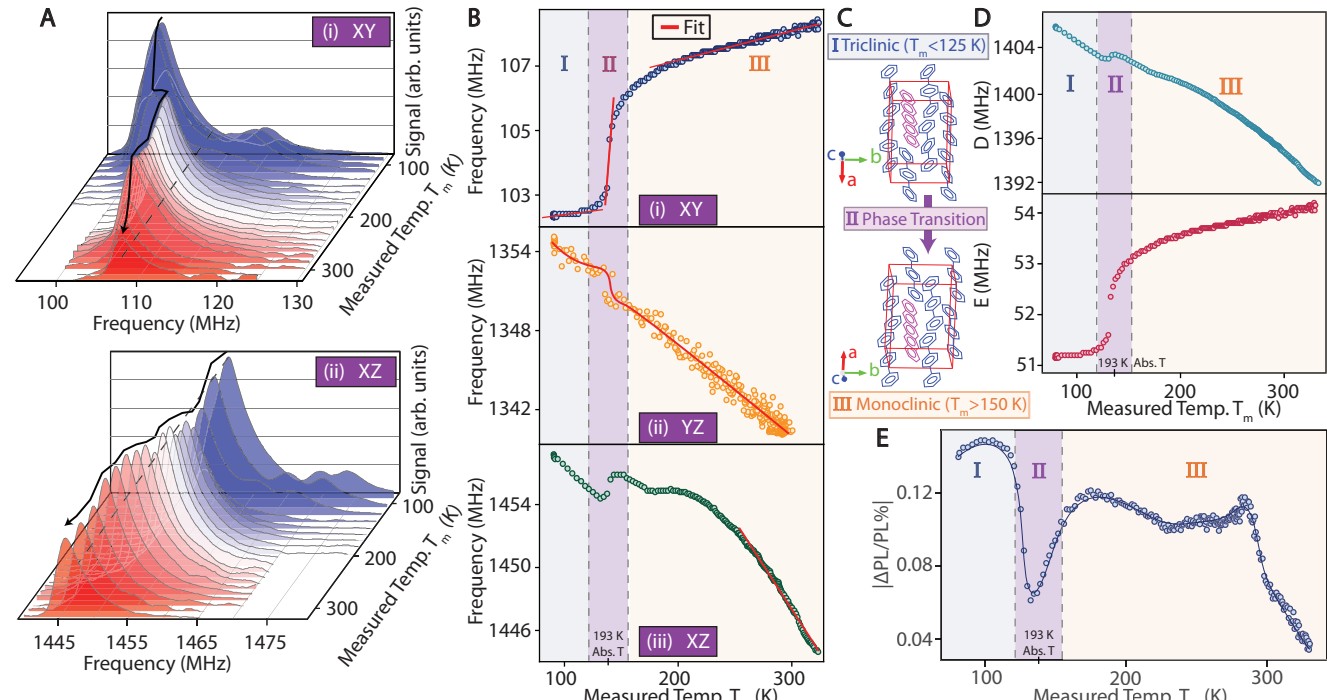

**Fig. 2 | Triplet ODMR temperature sensing. A** *ODMR spectra* of (i) $T_{xy}$ and (ii) $T_{xz}$ transitions with changing temperature. Reported are values from the cryostat cold-finger, and do not account for sample heating. Blue-to-red colors represent increasing temperature. Dashed line is guide to eye; black arrow tracks changes in spectral peak positions. **B** *Temperature variation* of ODMR peak position for the (i) $T_{xy}$, (ii) $T_{yz}$ and (iii) $T_{xz}$ transition over a wide range (79 to 330 K). Three linear regions are observed, marked **I-III**, with distinct slopes. Sharp variation in region **II** is due to a phase transition, at an absolute temperature (abs. T) of 193 K[48,49]

(marked). **C** *Lattice phases* corresponding to regions **I** and **III** are identified as triclinic and monoclinic, respectively. Phases and the phase transition are color coded consistently between plots. **D** *Temperature dependence* of the zero-field splitting parameters, $D$(T) and $E$(T). **E** *Absolute value of ODMR contrast* extracted over the temperature range for the $T_{xy}$ transition. General contrast increase is observed at lower temperatures, with sharp contrast variation near the phase transition, and a decrease for $T > 290$ K.

measured at the cryostat cold-finger and have a constant offset from the actual sample temperature due to laser induced heating. A dashed line parallel to the temperature axis serves as a visual guide, making the spectral shift immediately apparent. The movement of the peaks is indicated by the black arrows.

Figure 2B(i) shows the extracted ODMR peak positions for $T_{xy}$ transition estimated from the center of the steep spectral edge, plotted against the measured cold-finger temperature. The data reveal three distinct linear regions, labeled **I − III** and shaded for clarity. The strong variation around region **II**) has the characteristic signature of a phase transition in the p-terphenyl molecules[47,48]. From the literature, this phase transition occurs at 193 K[49]; this is marked in Fig. 2B for clarity.

Figure 2C schematically depicts the lattice diagrams of the two phases, transitioning from triclinic in region **I** to monoclinic in region **III**. The structures are similar, except for the central p-terphenyl benzene ring, which is out-of-plane in the triclinic phase. While related signatures have been observed previously in photoluminescence[50], to our knowledge, Fig. 2B marks the first time an ODMR measurement is carried out directly at a phase transition. Overall, Fig. 2B demonstrates that pentacene molecules are sensitive reporters to changes in the host lattice configuration.

The red lines in Fig. 2B(i) show linear fits to the ODMR variation in the three regions. The slope in region **II** is about three times that of the variation in diamond (see Table 1 for a detailed comparision). As this phase transition is reversible, it may serve as an excellent bias point for quantum sensing thermometry. We anticipate another sharp, albeit irreversible, phase transition at the melting point around 486 K (absolute temperature).

Figure 2B(ii, iii) shows corresponding variations for the $T_{yz}$ and $T_{xz}$ transitions respectively. The step variation at the phase transition is visible in both cases. $T_{yz}$ exhibits an approximately linear dependence over the entire temperature range studied, and constitutes a wider linear dynamic range than in NV-diamond[51]. From a practical perspective, the complementary use of the steep $T_{xy}$ transition and the linear $T_{yz}$ transition allows for both high sensitivity and a large dynamic range in temperature sensing within the same system.

Figure 2D presents the extracted changes in the ZFS parameters with temperature, $D$(T) and $E$(T). Table 1 compares these variations with other quantum sensing materials, including NV-diamond, silicon vacancies ($V_{Si}^-$) in silicon carbide (SiC) (shown are values for the excited state), and negatively charged boron vacancies ($V_B^-$) in hBN. The third column, $df/dT$, represents the change in spectral frequency with temperature. For pentacene, we observe a variation of 247 kHz/K for $T_{xy}$ in region **II** and 101 kHz/K for $T_{xz}$ in region **III** (red line in Fig. 2B(iii)), both steeper than the variation in NV-diamond.

Figure 2E now examines variations in ODMR contrast, focusing on the $T_{xy}$ transition (see Supplementary Note 2 for other transitions). Contrast increases slightly at lower temperatures but shows a sharp change near the phase transition in region **II**. Another drop occurs after the plateau past 290 K, likely due to exciton delocalization at elevated temperatures and increasing temperature decreasing the polarization lifetime and changing the intersystem crossing rates and triplet depopulation rates. Notably, much higher absolute contrast (up to 17%) can be achieved using pulsed laser excitation[41], and we expect similar contrast variations as shown in Fig. 2E even in this case.

To evaluate the time-normalized temperature sensitivity of our measurements, we use $\eta^T = \sigma\sqrt{\tau}/\frac{dS}{dT}$ (2)[52], where $\frac{dS}{dT}$ is the maximum

**Table 1 | Comparison of quantum sensor platforms for temperature and pressure sensing**

| Material | Linewidth (FWHM) | Max Contrast | $\frac{df}{dT}$ | $\eta^T$ | $\frac{df}{dP}$ | $\eta^P$ |
|---|---|---|---|---|---|---|
| | MHz | % | $\frac{kHz}{K}$ | $\frac{K}{\sqrt{Hz}}$ | $\frac{kHz}{Bar}$ | $\frac{Bar}{\sqrt{Hz}}$ |
| Pentacene | $4.8_{(XY)}$ | $16.8_{(XY)}$ [41] | $247_{(XY)}$ * | $0.04_{(XZ)}$ | $1800_{(XZ)}$ † | $0.07_{(XZ)}$ |
| $NV^-$ | $3.3$ [25] | $30$ [77] | $74.2$ [25] | $7.6 \times 10^{-4}$ [78] | $1.46$ [26] | $6$ [26] |
| $V_{Si}^-(SiC)$ [53] | $100$ | $0.11$ | $1100$ | $1$ | $0.031$ [79] | NR [79] |
| hBN [78] | $34$ [80] | $0.1$ | $684$ | $3.82$ | $91_{(\sigma_z)}$ | $26.2_{(\sigma_z)}$ |

Source references are shown as footnotes. Slopes of ODMR frequency variations $df/dT$ and $df/dP$ are material properties and reflect the intrinsic spin-environment interactions in the host material, provided that extrinsic perturbations (e.g., local heating, power broadening) are avoided. The signal strength and detection sensitivity ($\eta^T$, $\eta^P$) are affected by experimental parameters such as laser power, sample size, and RF/MW conditions. For pentacene, subscripts identify triplet transition from where values are extracted for sensitivity estimation. First two columns show ODMR linewidth and contrast but are not employed for sensitivity estimation.
*Taken from phase transition region. - For 1-2 bar. Values taken from highest performing regions. ref: [41]Singh et al. Phys. Rev. Research 7, 013192 (2025), measured under pulsed laser excitation.
[25]Acosta et al. Phys. Rev. Lett. 104, 070801 (2010), dD/dT cited for diamond. [77]Ho et al. Functional Diamond 1, 160 (2021),[78]Gottscholl et al. Nature Comm. 12, 4480 (2021),[26]Doherty et al. Phys. Rev. Lett. 112, 047601 (2014), dD/dP cited. [53]Kraus et al. Scientific Reports 4, 5303 (2014), dD/dP cited. [79]Wang et al. Nature Materials 22, 489 (2023), dD/dP cited. [80]Stern et al. Nature Comm. 13, 618 (2022).

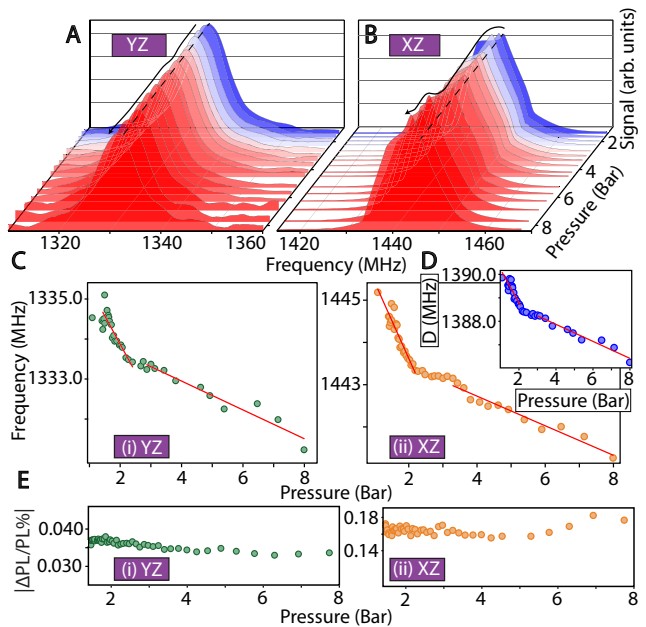

**Fig. 3 | Triplet ODMR isotropic pressure sensing. A** *Representative ODMR traces for the* (**A**) $T_{yz}$ *and* (**B**) $T_{xz}$ *transitions under varying isotropically applied pressure 0-8 bar. Colors blue-to-red indicate increasing pressure. Dashed line is guide to eye; black arrow tracks changes in spectral peak positions.* **C** *Pressure variation of the ODMR peak frequency for the* (i) $T_{yz}$ *and* (ii) $T_{xz}$ *transitions, both exhibit high sensitivity (see Table 1). Red lines are linear fits to the data.* **D** *Pressure dependence of ZFS parameter D(P) extracted from* (**C**). **E** *Variation in ODMR contrast for* (i) $T_{yz}$ *and* (ii) $T_{xz}$ *transitions. Contrast remains approximately constant over the range studied.*

parameters, such as photon collection and the number of spins interrogated. Nevertheless, even with our current setup, pentacene outperforms defects in SiC and hBN, partially due to the narrower ODMR linewidth[53] (first column in Table 1). With straightforward improvements, we anticipate approaching NV-diamond sensitivity; already however pentacene already offers significant deployment advantages due to the ease of crystal growth and lower cost (see Fig. 1C).

**Triplet ODMR variation with pressure**
An analogous study was conducted for ODMR variation with applied isotropic pressure, shown in Fig. 3 for a low absolute pressure range (0–8 bar). Figure 3A, B presents representative ODMR traces for the $T_{yz}$ and $T_{xz}$ transitions only, since the $T_{xy}$ transition is not as sensitive to pressure over the applied ranges studied (see Supplementary Fig. S4). The traces show minimal spectral broadening with applied pressure, along with a discernible shift in the peak position. In contrast to NV-diamond which is more suited to high-bias pressure environments, Fig. 3A, B demonstrates the ability for measurements at close to ambient conditions. Fig. 3

C shows the variation in ODMR transition frequencies for the (i) $T_{yz}$ and (ii) $T_{xz}$ transitions, similar to Fig. 2B. As expected, no lattice phase transition occurs within the pressure range studied. The variation in Fig. 3C is weakly nonlinear, but for simplicity, we estimate two linear slopes over the measured range, indicated by the red lines. For the $T_{xz}$ transition, we estimate a $df/dP$ variation of 1.8 MHz/Bar in the (1–2 Bar) range and 350 kHz/Bar in the (3–8 Bar) range. Variation for the $T_{yz}$ transition is similar: 1.4 MHz/Bar in the (1–2 Bar) range and 362 kHz/Bar in the (3–8 Bar) range.

As shown in the fifth column of Table 1, the maximum variation here is at least 1200 times greater than that of NV centers in diamond, and even larger for the case of $V_{Si}^-$ in SiC. This difference can be attributed to the relative weakness of the p-terphenyl lattice; also reflected in the lower Young's modulus in p-terphenyl (70 kBar[54]) compared to diamond (12 MBar) and SiC (4.5 MBar)[55]. Figure 3D presents the extracted $D(P)$ parameter, while $E(P)$ remains approximately constant over the range studied (see Supplementary Note 3, second paragraph)[49].

Figure 3E shows the variation in ODMR contrast over the pressure range, for the (i) $T_{yz}$ and (ii) $T_{xz}$ transitions respectively. The contrast is approximately constant throughout the range. Under the current conditions, time-normalized pressure sensitivity can be evaluated as $\eta^P = \sigma\sqrt{\tau}/\frac{dS}{dP}$ (3), and is reported in the sixth column in Table 1. Even without optimization, the pressure sensitivity for PDP ($\approx 0.07$ Bar/$\sqrt{Hz}$) significantly outperforms the best reported values for other platforms, while operating in a convenient range near ambient pressure.

ODMR signal slope with temperature, $\sigma$ is the noise floor, and $\tau$ is the integration time, defined by the low-pass filter's settling time in the detection lock-in amplifier. This definition incorporates both intrinsic properties (e.g., spectral shift) and extrinsic factors (e.g., noise level and integration time), and thus serves as a comprehensive figure of merit for comparing different experimental setups and material systems. Our setup is not optimized for sensitivity; we collect only a small fraction of photons, and the ODMR contrast in Fig. 2E is low due to continuous-wave illumination. Both factors could be improved by at least an order of magnitude[41].

Even so, in region **II** of the $T_{xz}$ transition (Fig. 2B(iii)), we estimate a sensitivity of $\eta^T = 0.04$ kHz/K. The fourth column of Table 1 shows the best reported values from other systems. A direct comparison is challenging, as sensitivity depends on many measurement

## Table 2 | Pentacene HOMO/LUMO Character from DFT (A) 3D renderings of the HOMO and LUMO molecular orbitals, indicated in yellow, in pentacene

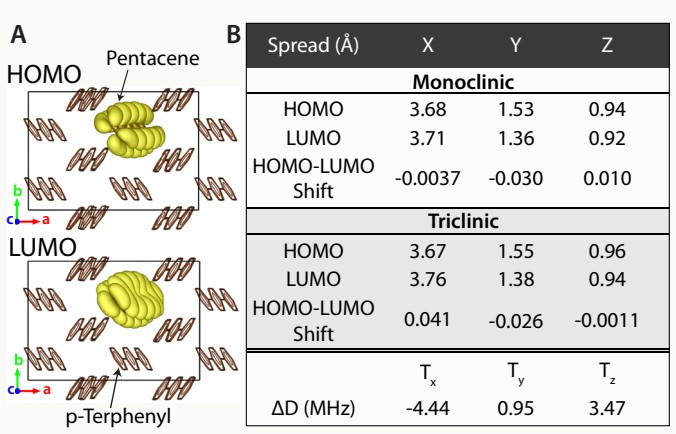

| Spread (Å) | X | Y | Z |
|---|---|---|---|
| **Monoclinic** | | | |
| HOMO | 3.68 | 1.53 | 0.94 |
| LUMO | 3.71 | 1.36 | 0.92 |
| HOMO-LUMO Shift | -0.0037 | -0.030 | 0.010 |
| **Triclinic** | | | |
| HOMO | 3.67 | 1.55 | 0.96 |
| LUMO | 3.76 | 1.38 | 0.94 |
| HOMO-LUMO Shift | 0.041 | -0.026 | -0.0011 |
| | $T_x$ | $T_y$ | $T_z$ |
| ΔD (MHz) | -4.44 | 0.95 | 3.47 |

(B) Pentacene HOMO and LUMO parameters in monoclinic and triclinic p-terphenyl phases, obtained from DFT calculations. HOMO/LUMO spread is the standard deviation of the pentacene HOMO/LUMO orbital density $|\phi|^2$. HOMO-LUMO shift is the difference of the centroid of the orbital densities. $\Delta\mathbf{D}$ is the change in zero field splitting tensor eigenvalues between the monoclinic and triclinic phases, for the $T_1$ exciton in the $\hat{\mathbf{x}}$, $\hat{\mathbf{y}}$, $\hat{\mathbf{z}}$ directions.

### DFT calculations

To understand these trends, we perform density functional theory calculations using a plane-wave basis set and norm-conserving pseudopotentials as implemented in the Quantum ESPRESSO code[56]. We use a kinetic energy cutoff of 60 Ry, and adopt spin collinear Perdew-Burke-Ernzerhof (PBE) as the exchange-correlation energy functional. Initial crystal structures of monoclinic and triclinic p-terphenyl were obtained from ref. 57. One pentacene molecule is substituted into the M4 position[58], and the ground state structures are further optimized until the Hellmann-Feynman forces on each atom are smaller than $10^{-5}$ au in magnitude. We use a $1\times1\times2$ supercell containing 1 pentacene molecule and 15 p-terphenyl molecules. Brillouin-zone integrations are sampled on a uniform grid of $2\times2\times1$ k-points. To simulate the optically excited triplet state, we follow the ΔSCF method[59], fixing the occupations so that the highest occupied molecular orbital (HOMO) of pentacene contains one spin-up electron and the lowest unoccupied molecular orbital (LUMO) of pentacene contains one spin-up electron. Renderings of these orbitals are shown in Table 2A.

We focus here on the restricted question of computing the change $\Delta D$ corresponding to the phase change between monoclinic and triclinic in Fig. 2B, D. Between the monoclinic and triclinic phases, we find relative shifts between the pentacene HOMO and LUMO centroid position as well as decreases in the spread of these orbitals. Orbital spreads are generally smaller in the monoclinic phase by 1 -5 pm (Table 2B). Lattice constants in the monoclinic phase are larger than the triclinic phase by about 0.2 Å in the $a$, $c$ directions, leading to reduced intermolecular interactions and tighter localization of molecular orbitals. Because of the crystal field, pentacene orbitals are not perfectly centered on the molecule. In particular, orbital centroids shift by 1 -4 pm between monoclinic and triclinic phases, with the largest relative shift along the long molecular axis ($\hat{\mathbf{x}}$).

These changes in the molecular orbitals lead to differences in the zero-field splitting tensor between the two phases. The spin-spin interaction Hamiltonian[60] is

$$\mathbf{D}_{ab} = \frac{1}{2}\frac{\mu_0}{4\pi}(g_e\mu_B)^2 \sum_{i<j} \chi_{ij} \left\langle \Phi_{ij} \left| \frac{r^2\delta_{ab} - 3r_a r_b}{r^5} \right| \Phi_{ij} \right\rangle \quad (1)$$

for all electron pairs $\Phi_{ij}(r, r') = \frac{1}{\sqrt{2}}(\phi_i(r)\phi_j(r') - \phi_j(r)\phi_i(r'))$ (5) and $\chi_{ij} = \pm 1$ for parallel( + )/anti-parallel( − ) electrons. Considering just the contribution from the pentacene HOMO and LUMO orbitals, we find differences in $\mathbf{D}_{ab}$ eigenvalues between the monoclinic and triclinic phases of up to 4 MHz (Table 2B), which is the same order of magnitude as the experimentally observed frequency shifts (Fig. 2B).

Furthermore, the $T_x$ frequency shifts the most, as a result of the larger orbital shifts along the $\hat{\mathbf{x}}$ direction, agreeing with the experimental observation of $T_{xy}$ and $T_{xz}$ transitions shifting the most. Including contributions of all 1392 electrons in the system to $\mathbf{D}_{ab}$ is computationally unfeasible. We expect these contributions to change the exact values of frequency shifts, while remaining at the same order of magnitude. We may estimate the change in the spin-spin interaction as $\Delta\mathbf{D} \approx \frac{1}{2}\frac{\mu_0}{4\pi}(g_e\mu_B)^2 \frac{\Delta r}{r^4}$ (6). Taking typical values from Table 2B of the change in localization length $\Delta r = 4$ pm, and the orbital spread $r \approx 3.7$ Å, we estimate $\Delta\mathbf{D} \approx 1$ MHz. This analysis shows that picometer scale changes in molecular orbitals are measurable by ODMR peak shifts at the MHz scale for such systems.

## Discussion

Our work suggests several intriguing future directions. As Table 1 highlights, pentacene-doped p-terphenyl crystals are compelling for pressure and temperature sensing. Other advantages, such as the ability to grow large (cm-scale) crystals (Fig. 1C) and easily cleave them, suggest the potential for large-area P, T sensor arrays. The crystals are free of paramagnetic impurities (unlike diamond P1 centers[61]) allowing intrinsically large sensor densities. Table 1 indicates that these materials are particularly suitable for high-temperature dynamic range and low-bias pressure environments, a complementary regime to diamond and SiC, which are better suited to high-bias pressure settings.

More broadly, our work highlights the benefits of chemical systems for P,T quantum sensing. This approach does not rely on electronic defects in semiconductor lattices and opens new design possibilities through chemical synthesis[31]. We anticipate increasing sensitivity by incorporating these molecules into porous materials such as metal-organic frameworks (MOFs)[37], where higher structural flexibility can result in greater sensitivity to pressure and strain. This also suggests new quantum sensor form factors, including thin films[40], 3D printed materials[62,63], and nanoparticles[64], possibly down to the single-molecule level[65]. The all-optical nature of our detection scheme offers a non-invasive route for local sensing, eliminating the need for physical contact or electrical interfaces for readout and anticipating single cell-deployable molecular temperature and strain sensing tags.

The large spin-strain coupling in these materials and their ease of fabrication significantly reduces the technical barrier to mechanically actuating the electronic spins[28], for instance via micromechanical structures. It also presents a novel pathway to linearly shift resonance frequencies of individual molecules via strain[66–68], suggesting a method to individually address qubits in molecular quantum computing and sensing platforms[69].

## Methods
### Sample preparation

The procedure to crystallize pentacene-doped p-terphenyl was adapted from Oxborrow et al.[70]. p-terphenyl (≥99.5% purity) and pentacene (99% purity) were purchased from Sigma-Aldrich. To further purify these raw materials, p-terphenyl was loaded into a borosilicate tube (I.D.: 8 mm, O.D.: 10 mm) and sealed under vacuum and inert atmosphere. The tube was then passed through > 30 rounds of zone refinement. Pentacene was purified via sublimation under continuous Ar (99.999% purity) supply in a borosilicate tube (I.D.: 8 mm, O.D.: 10 mm) in the dark to prevent light-induced disproportionation reactions[71,72]. Under inert atmosphere,

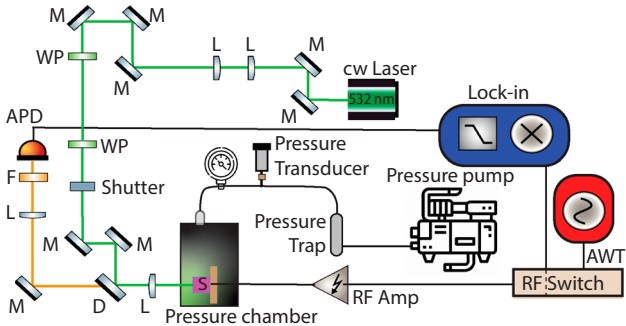

**Fig. 4 | Experimental setup for pressure sensing ODMR measurements.** The green line represents the laser beam from the cw laser, M represents the reflecting mirrors, L the convex lenses, and F the 600 nm long-pass filter. D is a 550nm long-pass dichroic mirror. The purple rectangle labeled S is the pentacene-doped p-terphenyl single crystal sample located inside a pressure chamber. Additional pressure instrumentation includes the gauge for monitoring, and the transducer, trap, and pump for maintaining pressure. Lock-in amplifier is used for detection and AWT for waveform generation.

a 1:1000 (w/w) ratio of purified pentacene and p-terphenyl was ground in a mortar and pestle via liquid-assisted grinding with a few drops of toluene and loaded into a homemade borosilicate glass crystal growth ampule (I.D.: 8 mm, O.D.: 10 mm). The ampule was flame-sealed under vacuum and inert atmosphere and loaded in the growth furnace. A preprogrammed stepper motor gradually lowered the ampule through a temperature gradient at 5 mm/hr to produce pentacene-doped p-terphenyl single crystals following the Bridgman method described in refs. 73,74.

### Further details of ODMR setup

Here, we provide additional details about the experimental setup used in our study. The experiments were conducted using a custom-built apparatus. For the temperature sensing experiments the sample was placed in a variable-temperature flow cryostat (Janis ST100) as shown in Fig. 1F. Figure 4 shows the setup used to record ODMR spectra for pressure sensing. The sample was placed in a pressure chamber, with pressure maintained using the pressure gauge, transducer, trap, and pump. A 532nm continuous wave laser (532nm 2W Coherent Verdi G2) was used for sample excitation. For the CW-ODMR experiment, the sample was continuously illuminated. The laser was reflected towards the sample with the help of a 550 nm long-pass dichroic mirror (D; Thorlabs DMLP550). The laser beam was focused on the sample at the final lens (focal length 50 mm), and the emitted photoluminescence (PL) was also collected using the same lens. Emitted PL was passed through the dichroic mirror, filtered with a 600 nm long-pass optical filter (F; Thorlabs FEL0600), and then focused on the avalanche photodiode (APD; Hamamatsu C12703 DC-10MHz) with the help of another lens. The signal from APD is recorded with a Lock-in amplifier (SRS model SR830 DSP). The laser power reaching the sample is ≈110 mW.

Radio frequency and microwave signals were generated using an arbitrary waveform transceiver (AWT) from Tabor Electronics (Model P9484M). We operated the Tabor AWT on NCO Mode. The generated RF or MW was passed through an RF-Switch (Mini-Circuits ZASWA2-50DR-FA+) before amplification with RF- and MW-Amplifier (Empower RF Systems 1079-BBM1C3K7G and Mini-Circuits ZHL-2W-63-S+, respectively). The RF was fed to the sample (S) through a 3 mm diameter 1-turn coil. The sample size is 2 mm × 2 mm × 0.9 mm and sits in the middle of the coil, which is terminated by a 50 Ohm resistor. The MW power reaching the sample was ≈1 W for pressure sensing experiments and ≈0.3 W for temperature sensing experiments.

### Contrast measurements

As mentioned above, the PL is collected using convex lenses and converted into an electrical signal using an APD module. This electrical signal is demodulated using a lock-in amplifier, whose reference signal modulates the RF (ON/OFF) at 1 kHz. The ODMR contrast is defined as the difference in the PL signal when the RF power is ON and OFF, divided by the PL signal when the RF is OFF. To express the ODMR contrast as a percentage, the calculated contrast is multiplied by 100[75].

## Data availability

The ODMR data generated in this study has been deposited in the Zenodo repository[76] and is available at (https://doi.org/10.5281/zenodo.17231876).

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

## Acknowledgements

We gratefully acknowledge discussions with J. Breeze, D. Marchiori, and S. Bhave, and funding from NSF TAQS, ONR (N00014-20-1-2806), AFOSR YIP (FA9550-23-1- 0106), the Noyce Foundation, and the CIFAR Azrieli Foundation (GS23-013). Work at the Molecular Foundry was supported by the Office of Science, Office of Basic Energy Sciences, of the U.S. Department of Energy under Contract No. DE-AC02-05CH11231. This research used resources of the National Energy Research Scientific Computing Center (NERSC), a Department of Energy Office of Science User Facility using NERSC award (FES-ERCAP0033471).

## Author contributions

H.S.—experiment conceptualization, design, and implementation, and data collection, processing, and analysis, and manuscript writing, revision, and submission N.D.—interpreted data, literature review, data visualization, manuscript writing, revision, and submission J.G.—data visualization, manuscript writing, revision, and submission A.S.—data visualization B.B.—supported experimental work E.D.—experiment design and implementation R.M.—grew samples, manuscript revision L.T.—DFT calculations, data visualization, manuscript writing, revision, and submission A.A.—funding acquisition, experiment design and implementation, data visualization and interpretation, manuscript writing, revision, and submission.

## Competing interests

The authors declare no competing interests.
