## [Transparent Peer Review file · Nature Communications]

High sensitivity pressure and temperature quantum sensing in pentacene-doped p-terphenyl single crystals

Corresponding Author: Professor Ashok Ajoy

Version 0:

Reviewer comments:

Reviewer #1

(Remarks to the Author)

Please, see attached file for my comments.

Reviewer #2

(Remarks to the Author)

In the submitted manuscript, the authors report on the utility of pentacene-doped in p-terphenyl host lattice as pressure and temperature sensor. The main idea behind the study is the leveraging of flexible nature of molecular crystals, relative to the well-investigated diamond lattice hosting NV centres, as a better sensor material. The manuscript is well-structured and fundamental concepts are eloquently presented. The authors need to address the following the comments to determine the suitability of the script for publication in Nature Communications.

Title:

The term organic crystals is rather general. I think, the title would be more informative, if the authors use "pentacene-doped p-terphenyl single crystals" instead of organic crystals.

Abstract:

Please mention temperature and pressure ranges studied.

I. Introduction

1. The authors cite references 28 and 29 as dealing with rare-earth ions. While it's true 29 deals with rare-earth ion (Eu³⁺), the reference 28 deals with Cr⁴⁺ molecules, a transition metal system. Please correct this discrepancy by rewriting the sentence. There is a typo: "utilizing are-earth ions"

2. What does the authors mean by tunable sensor placement in three-dimensions? Please elaborate.

3. For easy referencing, the authors can give values for the size of the single crystals and doping levels of sensor molecule in a host.

II. System and principle

Does the doping concentration affect the sensing capability of the system?

The figure 1E is labelled with 79 K, while the text says 77 K.

Please show the TXZ linewidth at 79 K (77 K?)

III. Triplet ODMR variation with temperature

In Figure 2C, the authors can add temperatures for easy referencing of temperature-dependent phase change.

Since pentacene performs a useful function, the authors can write "sensitive reporters" instead of "sensitive spectators"

In figure 1D and E, the ODMR frequencies of XY and XZ transitions increase with decreasing temperature. However, in figure 2A, for XY, the trend seems opposite. I am a bit confused how to interpret the data presented in the figures. Probably, I am missing a point here.

The attribution of drop occurring after 290 K to the crystals being close to the melting point (486 K, as mentioned by the authors) is rather tentative. The issue is compounded by almost 200 K difference. Could the authors provide a more logical explanation?

Apart from the frequency shift with respect to temperature, the intensity of the ODMR transitions seems to change with temperature. Could this intensity variation used to sense temperature, more like luminescence thermometry (<https://doi.org/10.1002/adma.202302749>) reported for luminescent molecular and solid-state systems? It would be

interesting, if the authors make a comparative study between intensity and frequency variation with respect to temperature.

IV. Triplet ODMR variation with pressure

Why the authors chose to study Tyz and Txz transitions? What is the rationale behind leaving out the Txy transition?

VI. Conclusions

I do agree with the authors that the studied system is interesting. However, I have difficulties accepting the system as a compelling pressure and temperature sensor. The same applies to the “elimination of reliance on electronic defects...” In my humble opinion, the statements are far-fetched and require rewriting.

Could the authors discuss how the chemical design strategies can be leveraged to improve the sensitivity of the crystalline system discussed in the manuscript? For example, could deuteration enhance the optical properties with the consequent effect on sensing capability?

A minor remark:

There is always a space between value and unit: for example, please correct 77-330K in the introduction, in the caption of Figure 1, and other parts of the manuscript.

Reviewer #3

(Remarks to the Author)

Singh et al. described a quantum sensing system based on pentacene in p-terphenyl crystals, which demonstrates high sensitivity to temperature and pressure through shifts observed in optically detected magnetic resonance (ODMR) spectra. While the study presents the novelty of using ODMR to measure these parameters, it is important to note that several key aspects of the work are already well documented in the literature. These include the use of pentacene triplet spin states for quantum sensing, the application of ODMR (and EDMR) to address these spin states, and the sensitivity of pentacene triplet states to both temperature, including phase changes, and pressure (e.g., ref 42, <https://doi.org/10.1021/jp070251v>).

Given that pentacene triplets are a well-established molecular platform, it is not surprising that the authors were able to optically detect spectral shifts and utilize them to sense these parameters. Nevertheless, I recognize that the use of emission-based detection in this context is a novel contribution, and I could potentially support the acceptance of the manuscript.

However, I would like the authors to address how they envision using this system as a sensor in real-world applications. Specifically:

- What types of target environments would this sensor be used for?
- Why are such capabilities necessary or advantageous in those contexts?

The authors claim that the pentacene triplet system exhibits higher sensitivity compared to defect-based systems such as nitrogen-vacancy (NV) centers. While this may be true, it is important to recognize that NV centers have been successfully implemented in both single-crystal and surface platforms for sensing these parameters in well-defined settings. The performance criteria listed in Table 1 have limited significance if they cannot be realized under practical conditions or if the system cannot be effectively deployed in its intended applications. For example, the authors highlight the phase transition region values for df/dT at 193 K to demonstrate a significantly larger sensitivity than NV centers. However, what is the practical relevance of this temperature in real-world sensing applications? A similar concern applies to pressure measurements—what is the significance of detecting variations in the 1–2 bar range? Without clear justification for these conditions, the claimed advantages of the pentacene system may not translate into meaningful practical benefits, and the presented results do not carry significant weight or novelty.

Overall, the methodology and data presented appear reasonably sound. I would recommend addressing these points to strengthen the paper's relevance and provide a clearer vision for its potential applications.

Minor comments:

p2: “The PDP crystal in Fig. 1C requires only \$2.25 in materials cost, representing a ~70,000-fold reduction in mass-normalized cost compared to NV-diamond.” This requires either citation or some support for the numbers.

p2: “Fig. 1D the case for RT and Earth's field.” Missing is?

Version 1:

Reviewer comments:

Reviewer #1

(Remarks to the Author)

Please, see attached file.

Reviewer #2

(Remarks to the Author)

The authors have satisfactorily responded to the criticism of the reviewers and revised the script according to the comments.

Therefore, publication of the script in Nature Communications is recommended.

Signed by

Senthil Kumar Kuppusamy

Reviewer #3

(Remarks to the Author)

The authors have conducted a careful revision, satisfactorily addressing my previous concerns. The manuscript now meets the scientific rigor and presentation standards required for publication in Nature Communications.

MANUSCRIPT NCOMMS-24-69634:
“High sensitivity pressure and temperature quantum sensing in organic crystals”
Referee Comments are in Black; Authors' Response are in Blue

RESPONSE TO REFEREE REPORT 1

The data presented are very clear, accurate and, overall, of good quality.

We thank the referee for their praise for the clarity, accuracy, and overall quality of the manuscript!

The comparison with the pressure and temperature sensing performed over different spin centers (such as – but not limited to - Nitrogen-Vacancy centers) allows us to appreciate the potential of molecular spins as sensors.

We are glad that the referee found the tabulated comparison between different pressure and temperature sensing platforms useful.

Notably and very interestingly, the final part of the manuscript contains DFT results which support the data and suggest the possible origin (at molecular orbital level) of the advantage of molecular spins for sensing pressure and temperature if compared to other magnetic spin impurities.

We thank the referee for appreciating the valuable theoretical underpinnings in our argument.

In my opinion the manuscript, in its actual version, is for sure reporting a molecular-based pressure and temperature quantum sensor, highlighting potential advantages with respect to other paramagnetic spin defects or impurities and a possible explanation about the origin of such advantages.

We thank the referee for their favorable assessment of the manuscript, the potential of the molecular pressure and temperature sensing platform presented, and fundamental work to understand the system's properties.

Overall, the results meet the level required for a publication in Nature Communications. However, at the same time, I have several points the authors must necessarily address before recommending the manuscript for publication. Please find below additional points and comments:

We are grateful to the referee for deciding that the revised manuscript meets the high standards of Nature Communications and for their recommendation to publish the revised manuscript.

1) References. In my opinion there are some missing references concerning sensing or quantum sensing of pressure and temperature. For instance, Ref. (5) in the caption of Table I, although mentioned in the table and used for comparison, is not appearing in the reference list. The same seems to occur also for Ref. (6), (8) and (9) of the same Table I. These works should be included in the reference list and properly cited (in the introduction and/or in the discussion part) since the authors are basing on their results to show the potential of their molecular sensor.

Here's possible other works the authors could consider to cite:

- <https://www.nature.com/articles/s41467-024-52272-y>

- <https://www.tandfonline.com/doi/full/10.1080/26941112.2021.1964926>

We appreciate the referee's suggestion of these additional references. We have now updated the manuscript to include these works in the main reference list and included the references of Ho et al. and Wang et. al. The formatting issues with Table I references have also been corrected.

It seems also that Ref [33] of the manuscript has been recently published and it is no more a preprint: <https://journals.aps.org/prl/abstract/10.1103/PhysRevLett.133.120801>

We have updated the manuscript to include the published version in the reference list.

The introduction mentions the potential of molecular spins systems for quantum sensing. In my opinion, the authors are missing some recent developments on both the theoretical as well as the experimental side, which should be added to the references:

- <https://iopscience.iop.org/article/10.1088/2058-9565/ad985e>
- <https://www.nature.com/articles/s41534-024-00838-5>
- <https://www.sciencedirect.com/science/article/abs/pii/S0304885319311199?via%3Dihub>
- <https://www.nature.com/articles/s41565-024-01724-z>
- <https://pubs.acs.org/doi/10.1021/jacs.2c07692>
- <https://pubs.aip.org/aip/jcp/article/158/16/161103/2885321/Toward-quantum-sensing-of-Chiral-induced-spin>

We appreciate the referee's recommendation of these additional quantum sensing references and believe their incorporation strengthens the context for and content of this work. We have drawn inspiration from these papers and have updated our introductory references to include them.

2) Doping level. The authors mention that the doping level of the crystal can be eventually high (introduction). However, the concentration used in the experiments is 0.1% corresponding to a collection of 10^9 spins and a volume of $2.6 \cdot 10^{-5} \text{ mm}^3$ (Section II). This is giving FWHM linewidth of $\approx 4.8 \text{ MHz}$. This concentration value is not so small and even so large. Could the authors comment on how the linewidth is affected by the concentration?

We thank the referee for the question. The ODMR lineshape exhibits a distinctive asymmetric profile, which we addressed in a previous manuscript [1]. The linewidth is primarily governed by hyperfine interactions with proximal ^1H nuclear spins in the pentacene aromatic rings [1]. While increased linewidth at higher concentrations is expected due to inter-electron interactions [2], the behavior is more nuanced. Several competing processes—such as singlet fission and triplet-triplet annihilation—contribute to the dynamics and are typically described using a Jablonski diagram that captures the relevant transition rates (see Ref. [3] for further discussion).

We also note a synthetic limitation in the achievable doping concentration for the p-terphenyl host, where it is difficult to exceed $\sim 3\%$ pentacene loading. In alternative host materials, however, higher concentrations may be feasible. We plan to explore these possibilities in future work and have expanded the discussion and references in the revised manuscript to clarify these points.

And on how much the concentration could be, in principle, increased or further decreased?

Crystal preparation constrains the theoretical maximum and minimum pentacene concentrations we can achieve. Experimentally, we have prepared doped crystals with concentrations up to 0.3% by mass (Fig. 1). Although higher concentrations are in principle possible for this system, a comprehensive study of the effects of such high concentrations has not yet been conducted (to the best of our knowledge), excluding the results described in [3] on thin films. Pentacene may likely have increased solubility in the crystal melt when doped into different host systems, resulting in a higher concentration in the final crystal. We are actively investigating the use of other hosts to produce different solid solution crystalline materials.

Lower pentacene concentrations are achieved by diluting the starting concentration of the pentacene/p-terphenyl

Figure 1: PDP crystals synthesized with varying pentacene concentrations. From left to right: 0.3%, 0.1%, and 0.05% pentacene doping, highlighting precise control over dopant levels and the ability to tune concentration across a broad range.

powder. The lowest concentration we have explored thus far is 0.025% by mass, but we anticipate that lower concentrations can be achieved.

3) Sample size. The sample shown in Fig. 1.c has a remarkably large size for a single crystal. However, the fraction of the volume used for experiments is only $2.6 \cdot 10^{-5} \text{ mm}^3$ which means that most of the volume available is effectively not necessary and not used during experiments. The same holds by considering the sample volume reported in Supplementary Material (Experimental Setup section). Which are the factors limiting the use of a larger sample volume?

We thank the referee for the thoughtful question. Our proof-of-concept experiments were deliberately conducted on a relatively small ensemble of spins, guided by two key motivations. First, our primary goal was to characterize the intrinsic sensitivity of the PDP material to environmental variables—specifically, the shifts in ODMR frequency with temperature and pressure, denoted $\partial f/\partial T$ and $\partial f/\partial P$. These parameters reflect the material's inherent responsiveness to thermal and mechanical stimuli and are largely independent of experimental conditions such as photon collection efficiency or instrumentation. Our measurements indicate that these shifts are significantly larger than those observed in NV centers in diamond.

Second, we chose to demonstrate these effects using cw laser excitation, aiming for practical deployment. Since PDP crystals are inexpensive to synthesize, we sought to show that both the materials and the measurement infrastructure can remain low-cost.

Using larger spin ensembles would indeed improve absolute sensitivity. Prior work in dynamic nuclear polarization (DNP) [4,5] and pentacene-based masers [6] has leveraged large para-terphenyl and naphthalene crystals for this purpose. However, these experiments typically require pulsed laser excitation to illuminate larger volumes and must consider depth-dependent absorption (Ref. [7] identifies $\sim 589 \text{ nm}$ as optimal). Adapting such strategies could yield substantial sensitivity gains for bulk pressure and temperature sensing.

How does the RF/MW field inhomogeneity affect the FWHM linewidth?

The referee raises a valid concern regarding RF inhomogeneity. RF/MW field inhomogeneity will increase the experimental FWHM linewidths. In the setup used in this work, however, the RF field is effectively uniform across the probed spin ensemble. This is achieved through the use of a solenoidal coil geometry, which confines the detection region well within the zone of RF homogeneity. Additionally, the design of microwave coils or resonators with large homogeneous volumes is well established—even for centimeter-scale crystals—as demonstrated in DNP and maser experiments. Instead, the dominant contribution to the observed linewidth is from hyperfine interactions with protons in the host material. This broadening can be reduced by deuteration [8].

4) Cost reported for sample in Fig. 1.c. The cost for the large crystal is estimated in 2.25 US dollars accounting only for materials. I really appreciate the comparison in terms of cost of production between a synthetic crystal and an artificial diamond (such as NV centers). However, in my opinion the comparison could be further extended and detailed. Possible points of interest for considering the cost might be:

- artificial diamonds are grown by chemical vapour deposition or by high-pressure high-temperature methods. Nitrogen and vacancies can be then created by ion implantation or neutron irradiation followed by thermal annealing. This initial difficulty can be overcome considering that there are specialized companies offering commercial NV centers.
- molecular spins offer an easier way of fabrication due to chemical synthesis. Here, however, raw materials with adequate quality and purity might be necessary and specific chemical techniques and chemical facilities for purification or making reactions. The recipe could require several complex steps and could be not so easy to be obtained or optimized at the beginning. The yield of the reaction could be not so high. Normalizing the cost over the mass of the product might be misleading as well. An alternative could be to

normalize over the density of the sample or on the spin concentration obtained (spins centers per unit of volume). This extended discussion could be added in a dedicated section of the Supplementary Material.

We thank the referee for their commendation of our cost comparison between the PDP crystal and NV-diamond materials. Please see the following breakdown for how we arrived at our conclusions from a materials and preparation angle. We have also updated the Supplemental Information to include a section on “Cost Analysis” for interested readers. Note that we have updated the discussion to normalize by spin density, which is a more relevant metric in this case.

Cost Analysis

Here, we detail how we arrived at the \$2.06 materials cost per crystal quoted in the main manuscript. Note that all prices quoted are current as of 4/22/25.

Materials Costs

For a typical PDP crystal, ≈ 1 mg of pentacene is doped into ≈ 1 g of p-terphenyl. Referencing Sigma Aldrich prices of \$179 for a 100 gram stock bottle of p-terphenyl and \$1350 for a 5 gram stock of pentacene, this comes to \$2.06 per ≈ 1 g crystal sample (\$0.00206 per mg or \$0.00000249 per ppb pentacene). NV-diamond samples are typically purchased from specialized commercial entities, eliminating the need for in-house high cost instrumentation for chemical vapor deposition (CVD), high-pressure high-temperature (HPHT) growth, and beamline work. A representative ThorLabs-supplied, Element Six-fabricated commercial DNVB1 single crystal NV-diamond sample has 300 ppb NV- concentration and is sold at \$1542.24 (\$97.92 per mg or \$5.14 per ppb NV-). The highest concentration DNVB14 diamond sample they supply has a 4.5 ppm NV- concentration, costing \$3325.46 (\$211.14 per mg or \$0.74 per ppb NV-).

Fixed Costs Associated with Growth

p-terphenyl and pentacene are purified in-house prior to the crystal growth, so it is useful to consider how this extra purification step contributes in the cost analysis. P-terphenyl is purified via zone-refinement. The setup we use employs nichrome wire, a 24 V power supply, glass tubing, a mechanical shuttling system (actuator and motor), and a motion control programmer. The primary costs here are for the power supply, mechanical actuator, and motion programmer, which cost \$52, \approx \$270, and \$46, respectively.

Figure 2: Homebuilt apparatus for high-throughput growth of large PDP crystals. Each modular setup—comprising a custom-built furnace and motor assembly—enables the growth of multiple centimeter-scale PDP crystals within 24 hours. Five such devices, designed for parallel operation, are shown.

Pentacene is purified via sublimation, accomplished in a home-built horizontal furnace. This setup consists of wire, a 24 V power supply, glass tubes, and high purity Argon gas. Presuming excess wire and glass tubes remain from the zone refinement setup construction, the primary cost of this step are the Ar gas cylinder and the power supply: \$66 and \$52, respectively.

Bridgman crystal growth is done in a home built furnace which uses wire, a ceramic tube, a 24 V power supply, a mechanical shuttling system, and a motion control programmer. The primary costs here are the power supply, shuttling system, and the motion control, \$52, ≈\$270, and \$46 respectively. The total cost, inclusive of preparation and crystal growth supplies, is a fixed ≈\$900 addition to the overall material price. Once multiple crystals are produced, the additional cost per crystal is just the material cost ≈\$2.

Summary

From a purely materials standpoint, even without normalization, buying the two stock chemicals to make a PDP crystal is cheaper than buying the cheapest ThorLabs-supplied Element Six diamond sample (\$1529 vs. \$1542). Normalization by mass (\$0.00206 per mg PDP vs. \$97.92 per mg or \$211.14 per mg NV-diamond, depending on the sample) OR by spin density (\$0.00000249 per ppb pentacene vs. \$5.14 per ppb NV⁻ or \$0.74 per ppm NV, depending on the sample) further highlights the cost benefits of using PDP crystals, relative to NV centers. The additional cost for the two purification and growth setups is a one-time, initial ≈\$900 investment (≈\$350 per step) which is still cheaper to produce at scale than commercial chemical vapor deposition instrumentation for diamond fabrication. For example, we have developed an in-house scale crystal production setup capable of mass-producing samples (**Fig. 2**). With all sample production and material costs taken into consideration, we conclude that PDP crystals are a significantly cheaper alternative relative to NV⁻ diamonds.

It seems that there is a slight broadening of the signals as the temperature is increased from 100 K to 300 K. However, Fig. 1.D and 1.E suggest the broadening of the lines occurs at low temperature, not at 300 K. Can the authors comment on this point?

We apologize for the confusion; the 3D visualization was used to make the shift more visually apparent and appealing in Fig. 2. However, we now realize it may have a deceiving effect on the eye for panel A. To aid in interpreting this data, we unambiguously show the frequency shift as a function of measured temperature in Fig. 2B. There is a consistent line broadening with decreasing temperature.

6) Section III, the change in the spectral frequency df/dT (and its equivalent for pressure sensing df/dP). It is stated that this quantity is material-specific and independent by the measurement apparatus or light collection efficiency. The temperature (or the pressure) is changed in the environment (cryostat or pressure cell) it is a global macroscopic experimental parameter of the whole sample (not a local one). Moreover, the setup it is said to be not optimized for sensitivity and that the collection efficiency affects the measured signal. In my opinion, the sample size (number of spins), the laser power, the RF/MW power can affect the absolute signal measured. So, one might expect df/dT (or df/dP) to be affected by the setup and experimental conditions as well. Could the authors better clarify this point?

The resonance frequency—if measured with sufficient precision—is governed by the intrinsic interaction between the spin system and its environment (temperature or pressure). Temperature- or pressure-induced changes in the spin Hamiltonian (e.g., via lattice strain, crystal field effects, or spin-phonon coupling) shift the resonant frequency, not the signal amplitude or quality of measurement. In this sense, the slope df/dT (or df/dP) is expected to be independent of the measurement apparatus under ideal conditions,

In this work, laser and RF/MW power were used within their linear regimes, and global temperature/pressure changes were well-controlled across the sample during measurement to minimize or eliminate drift-associated changes to the resonance frequency. To better clarify this point, we have revised the Table I caption in the main paper where this was stated to read:

"Slopes of ODMR frequency variations df/dT and df/dP are material properties and reflect the intrinsic spin–environment interactions in the host material, provided that extrinsic perturbations (e.g., local heating, power broadening) are avoided. The signal strength and detection sensitivity (η_T , η_P) are affected by experimental parameters such as laser power, sample size, and RF/MW conditions."

7) End of the Section III. It is mentioned that the sensitivity $\eta = \sigma\sqrt{\tau}/dT$ depends on many experimental parameters, including number of spins and collection efficiency so, it ultimately depends by the apparatus and experimental conditions. I agree with the definition of sensitivity used by the authors in the manuscript (which includes also the change in the spectral frequency). In my opinion introducing the noise floor level and the integration time with the mentioned definition of sensitivity can give a better comparison among different experiments and, eventually, setups and samples. This is because it can be a way to normalize the change in the spectral frequency over different experiments and apparatus. Could the authors comment on this point?

We agree that sensitivity—defined as $\eta = \sigma\sqrt{\tau}/(df/dT)$ —is not solely a material property but is strongly influenced by the experimental setup, including factors such as collection efficiency, number of spins, and noise characteristics. Indeed, the inclusion of the noise floor σ and integration time τ in the definitions of temperature and pressure sensitivity we provided gives a complete and practical description of sensor performance. This form allows for a direct comparison between different experiments and platforms through normalizing by measurable, apparatus-dependent parameters.

We have updated the manuscript to explicitly state this:

“This definition incorporates both intrinsic properties (e.g., spectral shift) and extrinsic factors (e.g., noise level and integration time), and thus serves as a comprehensive figure of merit for comparing different experimental setups and material systems.”

We have also updated the “Sensitivity Measurement” section of the SI to now report the estimated values of σ and τ for our measurements, and we emphasize that optimization of these parameters—along with improvements in collection efficiency and spin ensemble size—could significantly improve the overall sensitivity.

In my opinion the reader would benefit from having more details about the available experimental parameters This could be part of an extended Table I in Supplementary Material.

We note that an apples-to-apples comparison between sensor platforms is challenging, given that the papers referenced for other sensor platforms did not always provide all the experimental details mentioned. With this in mind, we have expanded the platform comparison table in the Supplementary Information of the paper to include laser power, collection efficiency, and spin density (ppm) as reported. Parameters not explicitly reported were calculated to give reasonable estimates for purposes of evaluation.

9) Protocol used for quantum sensing. The manuscript is citing a preprint by almost the same group of authors (Ref. 34) in which quantum sensing of magnetic field is experimentally realized with a similar pentacene-doped crystal and using again an ODMR approach. The sensing protocol in Ref. 34 seems to be fully pulsed-wave in the sense that both optical pulses and RF/MW pulses are used, while in the present manuscript the optical excitation is given in continuous wave. Could the authors explain why they chose to use a continuous wave protocol instead of a fully pulsed-wave one? Is there any advantage in running the sensor for pressure or temperature this way? In my opinion, running the sensor in pulsed-wave and exploiting its quantum coherence (as type II sensor according to Ref. [1]) would give more convincing evidence for a quantum sensing protocol (not only a larger contrast).

A CW protocol was employed instead of the pulsed protocol used previously (Ref 3), because we designed the experiments with an eye towards low-cost applications. A fully CW protocol facilitates the use of

commercial lasers, which can be obtained in cheap, compact form factors. For example, the laser powers employed in these measurements are ~16 mW, a range easily covered by commercially available 532nm diode lasers. Electronics in the 1300-1500 MHz microwave frequency ranges employed are already commercially available, and the material itself costs only ~\$2 to make (see response to Referee 1 Question 4).

In total, the cost-effectiveness of individual technological components used here present a readily scalable pressure / temperature sensing platform. As noted previously, we emphasize that even with a simple CW scheme, we achieve record pressure sensitivities. The high sensitivity is intrinsic to the material used and is independent of the high SNR a pulsed protocol would provide a more commercially relevant result.

10) Experimental setup. The experimental setup is described in detail. However, there is no information on the optical power reaching the sample or on the RF/MW ones.

The reader might benefit in such additional technical information about the protocols and on how the excitations are given and which is their timing or duration (see also the introduction and point 9 above). Adding details about the role of the lock-in and on how the contrast is extracted from experimental data might help the audience which is not familiar with ODMR. This information could be added in Supplementary Material.

We thank the referee for pointing out where the description of our experimental setup could benefit from additional detail. We discuss the relevant experimental parameters here and have added information to the SI. For all experiments, we can only report the laser power at the optical window (either of the cryostat or the pressure chamber) because the laser power meter head is too large to fit inside the cryostat or pressure chamber itself. The laser power reaching the optical window was 110 mW and the MW power fed through the single loop resonator was ~ 1 W for pressure sensing experiments and ~0.3 W for temperature sensing experiments. We have updated the SI with this experimental information.

The PL is collected using convex lenses and converted into an electrical signal using an Avalanche Photodiode (APD) module (Hamamatsu, C12703). The electrical signal is demodulated with a lock-in amplifier (SRS model SR830 DSP), whose reference signal modulates the RF (ON/OFF) at 1 kHz. ODMR contrast is defined as the change in the PL signal when the RF power is ON and OFF, divided by the PL signal when the RF is OFF. We have updated the SI with this experimental information.

For more detail about the ODMR measurements, Figure 1 has been updated with the cw-sensing protocol employed and the SI was updated with a brief description of how contrast is calculated in the “Contrast Measurement” section. The same detection scheme was used for all experiments.

11) It could be helpful to add a guide for eyes also in Fig. 3.A and 3.B, in a way similar to what was done in Fig. 2.A and 2.B.

We appreciate the referee’s suggestion and have added a guide to the eye for clarity in Fig. 3A and 3B.

RESPONSE TO REFEREE REPORT 2

In the submitted manuscript, the authors report on the utility of pentacene-doped in p-terphenyl host lattice as pressure and temperature sensor. The main idea behind the study is the leveraging of flexible nature of molecular crystals, relative to the well-investigated diamond lattice hosting NV centres, as a better sensor material. The manuscript is well-structured and fundamental concepts are eloquently presented. The authors need to address the following comments to determine the suitability of the script for publication in Nature Communications.

We sincerely thank the reviewer for their thoughtful evaluation and positive remarks on the structure, eloquence, and clarity of our manuscript. We appreciate the acknowledgment of our study's central idea—

utilizing the flexible nature of molecular crystals, specifically pentacene-doped p-terphenyl, as a promising alternative to rigid diamond lattices for pressure and temperature sensing. We have carefully considered all the reviewer's comments and addressed them point by point in the revised manuscript and the accompanying response document.

Title: The term organic crystals is rather general. I think the title would be more informative, if the authors use "pentacene-doped p-terphenyl single crystals" instead of organic crystals.

We agree that specifying the material used can make the title more informative and precise. Accordingly, we have revised the title to: "High sensitivity pressure and temperature quantum sensing in pentacene-doped p-terphenyl single crystals"

Abstract: Please mention temperature and pressure ranges studied.

We have revised the abstract to include the specific temperature and pressure ranges studied in our experiments for improved clarity and context for the reader.

I. Introduction

1. The authors cite references 28 and 29 as dealing with rare-earth ions. While it's true 29 deals with rare-earth ion (Eu³⁺), the reference 28 deals with Cr⁴⁺ molecules, a transition metal system. Please correct this discrepancy by rewriting the sentence. There is a typo: "utilizing are-earth ions"

We apologize for the typo and have corrected this line in the introduction to "utilizing rare-earth or transition-metal ions".

2. What does the authors mean by tunable sensor placement in three-dimensions? Please elaborate.

We apologize if our phrasing caused any confusion, and we clarify here. In the phrase "tunable sensor placement in three-dimensions", we are referring to molecular quantum sensors used as linkers in metal-organic frameworks (MOFs) [9]. These crystalline porous materials are highly modular in their construction, so changing the identity of the linker or the metal ion / metal cluster node can adjust the sensor's spatial configuration through altering the MOF structure [10]. For added clarity in the introduction, we have changed the wording in the following text:

"These systems offer advantages stemming from bottom-up synthesis, tunable sensor placement in three-dimensions via integration into porous materials and molecular-level control over sensor properties".

3. For easy referencing, the authors can give values for the size of the single crystals and doping levels of sensor molecule in a host.

We have included the single crystal size and doping levels in the following phrase in the introduction: "Additionally, the material can be grown into large single crystals (3 cm) with high doping levels (~1000 ppm)".

II. System and principle

Does the doping concentration affect the sensing capability of the system?

Additional mechanisms set in at higher doping concentrations which make it challenging to unambiguously predict sensing capabilities at higher doping levels. We discuss contributing factors below.

The doping level of 0.1% (w/w) (~827 ppm) pentacene achieved here is high, compared to the record in similar quantum sensing materials. For example, semiconductor defects like the NV- center have achieved up to ~16 ppm [11]. As discussed in our response to Referee 1 Question 2, we have grown crystals with dopings up to 0.3% (w/w) thus far (see Fig. 1 of that same response), with the upper % doping limit set by pentacene's solubility in the p-terphenyl melt. Pentacene may have a higher solubility in other host materials.

In our current samples, ODMR linewidths are principally broadened by hyperfine coupling to proximal nuclear spins. Since the pentacene is dilutely incorporated into the lattice (~1:1209 pentacene:p-terphenyl

molecules), we expect that electron–electron dipolar interactions contribute minimally to peak broadening, which would decrease sensing capability.

At high pentacene concentrations in PDP thin films, previous work demonstrated the onset of other triplet pathways, such as singlet fission and quintet state formation [3]. Furthermore, as the inter-pentacene distance decreases, we may expect a greater relaxation contribution from dipolar interactions.

In summary, the high concentration molecular quantum sensor regime is an exciting but not yet well explored parameter space. We expect that striking the balance between high sensor concentration and minimal feeding of other triplet pathways to be the optimal triplet molecular quantum sensor profile.

The figure 1E is labelled with 79 K, while the text says 77 K.

We thank the referee for pointing out this typo. The correct temperature is 79 K, and the manuscript has been updated accordingly.

Please show the TXZ linewidth at 79 K (77 K?)

Figure 1 has been updated to show the TXZ linewidth, 8.9 MHz.

III. Triplet ODMR variation with temperature

In Figure 2C, the authors can add temperatures for easy referencing of temperature-dependent phase change.

Figure 2C has been updated with the measured temperature corresponding to each para-terphenyl crystal structure.

Since pentacene performs a useful function, the authors can write “sensitive reporters” instead of “sensitive spectators”

We have incorporated the suggested change in the manuscript.

In figure 1D and E, the ODMR frequencies of XY and XZ transitions increase with decreasing temperature. However, in figure 2A, for XY, the trend seems opposite. I am a bit confused how to interpret the data presented in the figures.

There was a mistake made in Illustrator during the figure preparation, we sincerely apologize for this error. The correct trend is that the transition frequency *decreases* with *decreasing* temperature for the YZ and XZ transitions and *increases* with *decreasing* temperature for the XY transition, as shown in Figure 2B. We thank the referee for noticing this error. We have corrected the figure in the revised manuscript.

The attribution of drop occurring after 290 K to the crystals being close to the melting point (486 K, as mentioned by the authors) is rather tentative. The issue is compounded by almost 200 K difference. Could the authors provide a more logical explanation?

We thank the referee for their comment. In general, the polarization lifetime decreases with increasing temperature due to the Boltzmann distribution. Also, triplet population and depopulation rates will change with increasing temperature, likely shortening the triplet state lifetime. This may also be related to signal loss from exciton delocalization at elevated temperatures; exciton hopping barriers are typically in the range of 25 meV. These are more logical explanations for why contrast decreases in this regime of the plot, rather than approaching the melting point, which is much farther away in temperature. We have updated the manuscript accordingly to read the following:

“Another drop occurs after the plateau past 290 K, likely due to exciton delocalization at elevated temperatures and increasing temperature decreasing the polarization lifetime and changing the intersystem crossing rates and triplet depopulation rates.”

Apart from the frequency shift with respect to temperature, the intensity of the ODMR transitions seems to change with temperature. Could this intensity variation used to sense temperature, more like luminescence thermometry

Figure 3: Variation of photoluminescence intensity with temperature.

(<https://doi.org/10.1002/adma.202302749>) reported for luminescent molecular and solid-state systems? It would be interesting, if the authors make a comparative study between intensity and frequency variation with respect to temperature.

We thank the reviewer for their suggestions for developing this system as a multimodal sensor platform. As the intersystem crossing (ISC) rate decreases with decreasing temperature [12], a higher PL is observed at lower temperatures (see Fig. 3). This is because most of the pentacene molecules undergo spontaneous emission from the excited singlet state to the ground singlet state, rather than undergoing ISC into the triplet state.

In Fig. 3, we have plotted the PL vs. temperature graph below and estimated the relative thermal sensitivity, $S_r(T)=0.11\%$, using the expression:

$$S_r(T) = (1 / PL(T)) \times (dPL(T) / dT) \times 100\% = 0.11\%.$$

The deviations from linearity are not currently completely understood. We are continuing to explore these deviations and the potential causes, including that the response may simply be nonlinear. While there is not a large PL variation with temperature, it is an interesting angle that we will explore in the future.

IV. Triplet ODMR variation with pressure

Why the authors chose to study Tyz and Txz transitions? What is the rationale behind leaving out the Txy transition?

Interestingly, not all pentacene triplet transitions demonstrate the same pressure sensitivity. In particular, the T_{XY} transition is not as sensitive to pressure over the applied ranges studied as the other two transitions, T_{YZ} and T_{XZ}. We have included the plots of the T_{XY} transition frequency’s variation with pressure below (see Fig. 4) and in the SI, as we did not have space to include it in the main paper. Additionally, we have updated the following line in the manuscript to make this point clearer:

“Fig. 3A-B presents representative ODMR traces for the TYZ and TXZ transitions only, since the T_{XY} transition is not as sensitive to pressure over the applied ranges studied (see SI Fig. 4).”

Figure 4: Frequency variation of the Txy transition with applied pressure.

VI. Conclusions

I do agree with the authors that the studied system is interesting. However, I have difficulties accepting the system as a compelling pressure and temperature sensor.

In our humble opinion, the pentacene-doped para-terphenyl (PDP) system shows strong potential as a pressure and temperature sensor. Its high sensitivity coupled with low experimental costs and scalable production make it well-suited for practical applications, as detailed below.

As discussed in responses to Referee 1 Questions 4 and 9, the material costs to set up scale crystal production are modest. Each crystal costs approximately \$2 with a one-time infrastructure cost of \$900. The equipment needed to run experiments, a cw 532 nm laser and microwave electronics, are commercially available. Aside from the initial investment in crystal production and acquiring equipment, the barrier to adopting this technology is low.

We envision this system evolving into a deployable quantum sensor, akin to existing NV-diamond magnetometers [13]. Its low component cost and compact footprint support the development of a portable sensing platform. Both single crystal and thin film PDP form factors can be evaluated to determine which is better suited for such applications.

The PDP system's 1–2 bar optimal pressure range aligns with key use cases, including tire pressure monitoring, hydraulic fluid sensing, and altimetry—each requiring reliable detection of small pressure changes (see Page 13 for a more detailed discussion). The high sensitivity demonstrated by this organic crystal also points to a broader class of organic molecule-based pressure sensors. Since pressure sensitivity is largely governed by the Young's modulus of the host lattice—which can be higher in organic molecular systems than in semiconductor defect systems—we anticipate further development of such sensors. Organic molecules spatially arranged via incorporation into MOFs offer a promising direction for expanding this material class.

The same applies to the “elimination of reliance on electronic defects...” In my humble opinion, the statements are far-fetched and require rewriting.

The referee's point about the elimination of semiconductor defect materials is well taken, and we have rephrased the discussion and outlook phrase to highlight molecular quantum sensors as a complementary material platform to defects in semiconductors with the following language:

“This approach does not rely on electronic defects in semiconductor lattices and opens new design possibilities through chemical synthesis.”

Could the authors discuss how the chemical design strategies can be leveraged to improve the sensitivity of the crystalline system discussed in the manuscript? For example, could deuteration enhance the optical properties with the consequent effect on sensing capability?

Two chemical design strategies that can improve the sensitivity of the crystalline system employed are deuteration of the host pentacene molecule and utilization of a pentacene derivative, 6,13-diazapentacene. As demonstrated in the X-band EPR study by Sloop et. al. [14], deuteration of the pentacene host increases the triplet state lifetimes of the T_{+1} and T_{-1} states (the T_0 state is relatively unchanged) and slightly improves the polarization lifetime for the T_0 to T_{+1} transition and T_{-1} to T_{+1} transition (T_0 to T_{-1} is slightly worsened). While the population rates of the 1 and -1 sublevels are slightly decreased relative to fully protonated pentacene, the difference in the population rates is improved, which would increase polarization in this transition (Table 1). We anticipate that the cumulative effect of improvement in triplet state lifetimes, polarization lifetimes, and triplet population rates would improve the polarization and enhance the SNR of the measurements.

	Population rate	Spin-lattice decay rate ^(a)	Decay rate
Pentacene-d₁₆	$P_1 = 0.4 \pm 0.05$	$W_1 = 1.77 \times 10^4$	$k_1 = 7.52 \times 10^3$
	$P_0 = 0.16 \pm 0.02$	$W_2 = 2.86 \times 10^4$	$k_0 = 10^3$
	$P_{-1} = 0.44 \pm 0.05$	$W_3 = 7.52 \times 10^3$	$k_{-1} = 6.76 \times 10^3$
Pentacene-h₁₄	$P_1 = 0.44 \pm 0.05$	$W_1 = 1.85 \times 10^4$	$k_1 = 1.52 \times 10^4$
	$P_0 = 0.09 \pm 0.02$	$W_2 = 2.63 \times 10^4$	$k_0 = 10^3$
	$P_{-1} = 0.47 \pm 0.05$	$W_3 = 0.658 \times 10^4$	$k_{-1} = 1.39 \times 10^4$

Table 1: Triplet state parameter comparison between deuterated and protonated pentacene doped in para-terphenyl. Reproduced from Ref 14.

^(a) W_1 is the rate for $|0\rangle \leftrightarrow |+1\rangle$ transition, W_2 for $|0\rangle \leftrightarrow |-1\rangle$, and W_3 for $|-1\rangle \leftrightarrow |+1\rangle$.

The best fit of the kinetic parameters at 25°C (s^{-1}). The estimated errors for all the parameters except k_0 are $\pm 10\%$. The estimated error for k_0 is $\pm 50\%$.

Similarly, 6,13-diazapentacene-doped para-terphenyl has recently been shown as a high contrast quantum sensing material (**Fig. 5**) for magnetometry applications [15]. Since the primary temperature and pressure sensitivity properties we show here result from the p-terphenyl lattice phase transitions and compressibility, we anticipate pressure and temperature sensing would be possible with this doped crystal as well, again with higher contrast and improved measurement SNR.

Lastly, pentacene or its chemical-derivatives could be incorporated as a building block into porous crystalline materials [16], such as metal-organic frameworks, for tunable sensor placement in 3D porous materials [17].

A minor remark: There is always a space between value and unit: for example, please correct 77-330K in the introduction, in the caption of Figure 1, and other parts of the manuscript.

We thank the reviewer for this remark. We have carefully revised the manuscript and ensured that a space is

included between numerical values and units throughout the text, including in the Introduction, Fig. 1 caption, and all other relevant sections.

Figure 5: Chemically tuning the ODMR contrast (from Ref. 14)

RESPONSE TO REFEREE REPORT 3

Singh et al. described a quantum sensing system based on pentacene in *p*-terphenyl crystals, which demonstrates high sensitivity to temperature and pressure through shifts observed in optically detected magnetic resonance (ODMR) spectra. While the study presents the novelty of using ODMR to measure these parameters, it is important to note that several key aspects of the work are already well documented in the literature. These include the use of pentacene triplet spin states for quantum sensing, the application of ODMR (and EDMR) to address these spin states, and the sensitivity of pentacene triplet states to both temperature, including phase changes, and pressure (e.g., ref 42, <https://doi.org/10.1021/jp070251v>). Given that pentacene triplets are a well-established molecular platform, it is not surprising that the authors were able to optically detect spectral shifts and utilize them to sense these parameters. Nevertheless, I recognize that the use of emission-based detection in this context is a novel contribution, and I could potentially support the acceptance of the manuscript.

We thank the reviewer for their thoughtful comments and for recognizing the novelty of our room temperature, emission-based detection approach for quantum sensing using pentacene in *p*-terphenyl crystals.

The pressure ranges accessible with the pentacene-doped para-terphenyl material enable operation in a complementary pressure range to NV- diamond-based sensors, a benchmark quantum pressure sensing material. The organic *p*-terphenyl crystal lattice is softer than that of the more rigid diamond-based systems. Mechanical softness enables the lattice to respond more sensitively to lower pressure variations, facilitating operation in the low-pressure regime where diamond NV-center sensors are less effective or overly stiff. This expands the application range of quantum sensing technologies.

We have added the reference mentioned to our discussion of phase transitions in this material.

However, I would like the authors to address how they envision using this system as a sensor in real-world applications. Specifically:

- What types of target environments would this sensor be used for?
- Why are such capabilities necessary or advantageous in those contexts?

We thank the referee for raising this point. We envision a broad range of applications for PDP crystals across diverse domains. While our discussion here is focused more on pressure/strain sensing, similarly wide applications can be envisioned for temperature sensing.

Strain-mediated spin actuation: One particularly compelling application lies in the strain-mediated control of spin systems. Prior studies have shown that electronic spins, such as those in nitrogen-vacancy (NV) centers in diamond, can be coherently actuated using strain fields, including standing-wave strain patterns produced by H-bar resonators [18] or through diamond cantilever devices [19,20]. This spin-strain actuation has evolved into a major research area, including efforts to control spins using single phonons [21].

Figure 6: Preliminary work showing pentacene doped crystals grown as needle like structures.

However, the implementation of these ideas in diamond has been hampered by the material's inherently low strain sensitivity. Typically, stress levels on the order of MPa are required to achieve meaningful spin actuation [18], necessitating either high-strain cantilevers or advanced nanofabrication techniques directly in diamond—both of which present challenges.

In contrast, PDP crystals offer an attractive alternative. These crystals are significantly easier to fabricate. In new unpublished work, we have shown that pentacene-based crystals can be grown in needle-like morphologies, which are directly amenable to use as cantilevers (see Fig. 6). The inherent strain sensitivity roughly three orders of magnitude higher than that of diamond makes it possible to actuate spins at substantially lower strain and power thresholds.

Optical pressure and strain sensing devices: More broadly, we envision PDP crystals as general-purpose pressure and strain sensors potentially surpassing the capabilities of classical devices (e.g. strain gauges). These sensors may find applications in contexts such as pitot tubes and altimeters. Furthermore, PDP-based sensors may enable measurements in contexts where traditional strain gauges are ineffective—for example, in mapping high-resolution strain distributions within heterogeneous materials or small-scale structures.

From an engineering perspective, these crystals are highly compatible with low-cost diode laser systems and fiber-based optics. This compatibility paves the way for compact, integrated, and scalable pressure sensing platforms, akin to recent fiber-based magnetometers demonstrated in Ref. [13].

Functional materials via additive manufacturing: We foresee exciting directions in the development of functional materials by incorporating pentacene or its derivatives into polymers or 3D-printed structures. These composite materials would have built-in optical readout capabilities for strain, pressure, or temperature, accessible simply through laser illumination. We have previously demonstrated a similar approach with NV centers in diamond [16,22]: by embedding NV-nanodiamonds into 3D printing filament, we fabricated intricate structures via two-photon polymerization (TPP) with spatially distributed sensing capability.

Fig. 7 illustrates one such porous gyroid structure, where nanodiamonds (seen as red fluorescence) line the interior pore walls. (Here, green autofluorescence arises from the host resin.) We envision analogous structures incorporating PDP nanocrystals, which would offer the added benefits of simpler processing, enhanced sensitivity, and the potential for full-field, in-situ strain mapping. Such capabilities could greatly benefit the design of topologically inspired metamaterials and strain-resilient architectures—areas where current in-situ strain measurement techniques remain limited. We also highlight relevant work by Carlos

Portela's group at MIT [23], where optical spectroscopy was used to map strain fields in complex 3D-printed structures. In comparison, PDP-based approaches offer a potentially simpler, built-in readout mechanism with higher sensitivity.

Thin-film strain sensors and acoustic applications:

Pentacene-based thin films, patterned directly onto surfaces, represent another promising platform for strain sensing. Optical interrogation of these films could allow for spatially resolved strain mapping across a substrate. Applications may include acoustic or vibrational sensing [24], where such films serve as highly responsive coatings that transduce surface deformations—such as those from sound waves—into measurable optical signals.

Figure 7: Nanodiamond (ND) quantum sensors in 3D structures assembled via TPP. (A) SEM of diamond porous structures in a “gyroid” cross section. (B) Zoomed confocal image showing ND in the internal walls. Red fluorescence is from NDs while green is autofluorescence from the resin.

Bottom-up engineered nanomaterials: On the nanoscale, we envision using pentacene molecules as prototypes of modular building blocks in the construction of mechanically responsive materials—particularly in metal-organic frameworks (MOFs) [25]. The high strain sensitivity of PDPs as we demonstrated in this work, stems from their soft mechanical properties, including a low Young's modulus. In MOF-like systems, one could chemically tune both the mechanical response by varying the internal pore structure, linker rigidity, and other design parameters. This opens a powerful avenue for bottom-up design of smart materials with customizable strain sensitivity, expanding the toolbox beyond what is possible with conventional semiconductors or defect centers.

Biological sensing applications: Finally, we see significant opportunities for PDP nanocrystals in biological contexts. These crystals can be synthesized in nanocrystalline form and functionalized for incorporation into living systems. Recent work by Ishiwata and coworkers [26] demonstrated successful integration of such nanocrystals into cells, with low ODMR heterogeneity—thanks to less onerous requirements for mechanical ball milling.

Once incorporated, these crystals could serve as intracellular strain reporters. For example, forces applied to cells—via optical tweezers or during migration—could be monitored through the embedded PDP nanocrystals, enabling studies of cell mechanics, viscoelasticity, and intracellular rheology [27,28]. We are also exploring similar applications in plant systems. In unpublished work from our group (Fig. 8), diamond nanoparticles embedded in plant roots yielded ODMR readouts from different regions of the rhizosphere. We believe similar experiments with PDP nanocrystals are not only feasible but potentially advantageous, enabling detection of mechanical stress in root tissues in response to environmental stimuli such as drought or changes in soil composition. These measurements could shed new light on how mechanical cues influence plant development.

Figure 8: Nanodiamond incorporated into different parts of a root of a living plant and corresponding ODMR signals (unpublished).

In summary, PDP crystals, with their superior strain sensitivity, offer a compelling platform for a diverse range of pressure and strain sensing applications.

The performance criteria listed in Table 1 have limited significance if they cannot be realized under practical conditions... For example, the authors highlight the phase transition region values for

df/dT at 193 K to demonstrate a significantly larger sensitivity than NV centers. However, what is the practical relevance of this temperature in real-world sensing applications?

We refer to the more detailed response above. Regarding the practical relevance of sensitivity benchmarks like df/dT at 193 K, we highlighted this value to illustrate the intrinsic potential of the molecular system. While 193 K may not directly correspond to application-relevant temperatures, it reflects the upper limit of achievable sensitivity and highlights physical mechanisms that can be tuned to improve performance at room temperature and beyond—a direction we are actively pursuing.

DFT calculations also suggest that picometer-scale changes are detectable by the molecular orbitals, even outside the phase transition region. In other words, any event producing orbital-level changes at that scale would be detectable, regardless of proximity to a phase transition. We have edited the discussion section as follows:

“This also suggests new quantum sensor form factors, including thin-films, 3D printed materials, and nanoparticles, possibly down to the single-molecule level. The all-optical nature of our detection scheme offers a non-invasive route for local sensing, eliminating the need for physical contact or electrical interfaces for readout and anticipating single cell-deployable molecular temperature and strain sensing tags.”

A similar concern applies to pressure measurements—what is the significance of detecting variations in the 1–2 bar range? Without clear justification for these conditions, the claimed advantages of the pentacene system may not translate into meaningful practical benefits, and the presented results do not carry significant weight or novelty.

We refer to the more detailed answer provided above on potential applications. Pressure variations in the narrow 1–2 bar range are highly relevant in biological and microfluidic contexts, such as monitoring changes in osmotic pressure or intracellular vesicle dynamics [27]. When deployed as nanoparticles, PDP molecular crystal-based quantum sensors can be used for additively manufactured, bottom-up engineered quantum sensing materials [16] or functional metamaterials [23]. If PDP nanoparticles are surface functionalized for biological-compatibility [26], then they can be used for in-cell sensing of small, but biologically meaningful, variations in pressure and temperature.

Overall, the methodology and data presented appear reasonably sound. I would recommend addressing these points to strengthen the paper's relevance and provide a clearer vision for its potential applications.

We thank the reviewer for their favorable comments on our methodology and the quality of our data and for their useful feedback. We hope that the revised version of the manuscript now clearly outlines the scope of applications for this system and the significance and novelty of the results.

Minor comments:

p2: “The PDP crystal in Fig. 1C requires only \$2.25 in materials cost, representing a ~70,000-fold reduction in mass-normalized cost compared to NV-diamond.” This requires either citation or some support for the numbers

Referee 1 had a similar comment related to how we arrived at the stated crystal materials cost, so we refer the reviewer to our answer to Referee 1’s Question #4 and the associated new section of the SI.

p2: “Fig. 1D the case for RT and Earth’s field.” Missing is?

We thank the referee for catching this typo and have updated the manuscript accordingly.

REFERENCES

- [1] H. Singh, N. D’Souza, K. Zhong, E. Druga, J. Oshiro, B. Blankenship, R. Montis, J. A. Reimer, J. D. Breeze, and A. Ajoy, Room-temperature quantum sensing with photoexcited triplet electrons in organic crystals, *Physical Review Research* **7**, 013192 (2025).
- [2] C. Kittel and E. Abrahams, Dipolar Broadening of Magnetic Resonance Lines in Magnetically Diluted Crystals, *Phys. Rev.* **90**, 2 (1953).
- [3] D. Lubert-Perquel, E. Salvadori, M. Dyson, P. N. Stavrinou, R. Montis, H. Nagashima, Y. Kobori, S. Heutz, and C. W. Kay, Identifying triplet pathways in dilute pentacene films, *Nature Communications* **9**, 4222 (2018).
- [4] T. R. Eichhorn, B. van den Brandt, P. Hautle, A. Henstra, and W. T. Wenckebach, Dynamic nuclear polarisation via the integrated solid effect II: experiments on naphthalene-h 8 doped with pentacene-d 14, *Molecular Physics* **112**, 1773 (2014).
- [5] P. Hautle and W. T. Wenckebach, Creating high, portable proton polarization with photo-excited triplet dnp, *Journal of Magnetic Resonance Open* **20**, 100159 (2024).
- [6] M. Oxborrow, J. D. Breeze, and N. M. Alford, Room-temperature solid-state maser, *Nature* **488**, 353 (2012).
- [7] Y. Quan, N. Niketic, J. M. Steiner, T. R. Eichhorn, W. Tom Wenckebach, and P. Hautle, General theory of light propagation and triplet generation for studies of spin dynamics and triplet dynamic nuclear polarisation, *Molecular Physics* **121**, e2169025 (2023).
- [8] A. Brouwer, J. Köhler, E. J. Groenen, and J. Schmidt, ¹³C isotope effects for pentacene in p-terphenyl: High-resolution spectroscopy and single-spin detection, *The Journal of Chemical Physics* **105**, 2212 (1996).
- [9] J. M. Zadrozny, A. T. Gallagher, T. D. Harris, and D. E. Freedman, A porous array of clock qubits, *Journal of the American Chemical Society* **139**, 20 (2017).
- [10] Zadrozny, Joseph M., Audrey T. Gallagher, T. David Harris, and Danna E. Freedman. "A porous array of clock qubits." *Journal of the American Chemical Society* 139, no. 20 (2017): 7089-7094.
- [11] V. M. Acosta et al., Diamonds with a high density of nitrogen-vacancy centers for magnetometry applications, *Phys. Rev. B* **80**, 11 (2009).
- [12] R. Brown, J. Wrachtrup, M. Orrit, J. Bernard, and C. Von Borczyskowski, Kinetics of optically detected magnetic resonance of single molecules, *The Journal of Chemical Physics* **100**, 7182 (1994).
- [13] R. Patel et al., Subnanotesla magnetometry with a fiber-coupled diamond sensor, *Physical Review Applied* **14**, 044058 (2020).
- [14] D. J. Sloop, H.-L. Yu, T.-S. Lin, and S. Weissman, Electron spin echoes of a photoexcited triplet: Pentacene in p-terphenyl crystals, *The Journal of Chemical Physics* **75**, 3746 (1981).

- [15] S. K. Mann, A. Cowley-Semple, E. Bryan, Z. Huang, S. Heutz, M. Attwood, and S. L. Bayliss, Chemically Tuning Room Temperature Pulsed Optically Detected Magnetic Resonance, *Journal of the American Chemical Society* (2025).
- [16] B. W. Blankenship, Z. Jones, N. Zhao, H. Singh, A. Sarkar, R. Li, E. Suh, A. Chen, C. P. Grigoropoulos, and A. Ajoy, Complex Three-Dimensional Microscale Structures for Quantum Sensing Applications, *Nano Letters* **23**, 9272 (2023).
- [17] L. Sun et al., Room-temperature quantitative quantum sensing of lithium ions with a radical-embedded metal–organic framework, *Journal of the American Chemical Society* **144**, 19008 (2022).
- [18] E. MacQuarrie, T. Gosavi, N. Jungwirth, S. Bhave, and G. Fuchs, Mechanical spin control of nitrogen-vacancy centers in diamond, *Physical Review Letters* **111**, 227602 (2013).
- [19] P. Ovarthaiyapong, K. W. Lee, B. A. Myers, and A. C. B. Jayich, Dynamic strain-mediated coupling of a single diamond spin to a mechanical resonator, *Nature Communications* **5**, 4429 (2014).
- [20] J. Teissier, A. Barfuss, P. Appel, E. Neu, Maletinsky, and P., Strain coupling of a nitrogen-vacancy center spin to a diamond mechanical oscillator, *Physical Review Letters* **113**, 020503 (2014).
- [21] H. Wang and I. Lekavicius, Coupling spins to nanomechanical resonators: Toward quantum spin-mechanics, *Applied Physics Letters* **117**, (2020).
- [22] B. W. Blankenship, Y. Rho, Z. Jones, T. Meier, R. Li, E. Druga, H. Singh, X. Xia, A. Ajoy, and C. P. Grigoropoulos, Optically Detected Magnetic Resonance Imaging and Sensing Within Functionalized Additively Manufactured Microporous Structures, arXiv Preprint arXiv:2502.16434 (2025).
- [23] Y. Kai, S. Dhulipala, R. Sun, J. Lem, W. DeLima, T. Pezeril, and C. M. Portela, Dynamic diagnosis of metamaterials through laser-induced vibrational signatures, *Nature* **623**, 514 (2023).
- [24] Y. Wang, R. Zhou, Z. Liu, and B. Yan, A low-power CMOS wireless acoustic sensing platform for remote surveillance applications, *Sensors* **20**, 178 (2019).
- [25] K. Orihashi, A. Yamauchi, M. Inoue, B. Parmar, S. Fujiwara, N. Kimizuka, M. Asada, T. Nakamura, and N. Yanai, Radical qubits photo-generated in acene-based metal–organic frameworks, *Dalton Transactions* **53**, 872 (2024).
- [26] H. Ishiwata, J. Song, Y. Shigeno, K. Nishimura, and N. Yanai, Molecular Quantum Nanosensors Functioning in Living Cells, (2025).
- [27] A. Vian, M. Pochitaloff, S.-T. Yen, S. Kim, J. Pollock, Y. Liu, E. M. Sletten, and O. Campàs, In situ quantification of osmotic pressure within living embryonic tissues, *Nature Communications* **14**, 7023 (2023).
- [28] H. Knowles, *Simultaneous Nanoscale Rheology and Thermometry in Complex Systems Using Diamond Nanocrystals*, in *Quantum Nanophotonic Materials, Devices, and Systems 2023* (SPIE, 2023), p. PC126570A.

MANUSCRIPT NCOMMS-24-69634:
“High sensitivity pressure and temperature quantum sensing in organic crystals”

Referee Comments are in Black; Authors' Response are in Blue

RESPONSE TO REFEREE REPORT 1

I have carefully checked the revised version of the manuscript by H. Singh and coauthors as well as their reply to my points. In my opinion, the authors have addressed all my previous points and to have also followed some of my suggestions.

Therefore, according to my previous report and assessment, I can recommend the paper for publication in Nature Communications in its actual revised version.

However, as a final remark and advice for the future, I strongly suggest the author to fully report the

whole comments of a Reviewer when addressing them in their reply, without cutting or omitting parts of them. More specifically, the original numeration of part of my comments appeared to be wrong and it was a little hard to compare them with the authors' replies. This additional effort from

my side could have been easily avoided.

There are several reasons behind this advice:

- i) The peer review file will be made available in case of publication. It would be much and much better for the readers to be able to track the whole revision history (i.e., full comments and replies). No matters about the length.
- ii) This improves the readability of the revisions made by the authors
- iii) apparently, a similar thing was not done with the comments of the other Reviewers. This could be misunderstood for a lack of consideration towards one of the Reviewers
- iv) The authors should know that omitting part of the considerations and motivations given by a Reviewer in his/her report might somehow hide or mask his/her critical thinking on the manuscript.

Author's response: We sincerely thank the reviewer for the positive evaluation of our revised manuscript and for the recommendation for publication. We also appreciate the valuable advice regarding the inclusion of full reviewer comments in our future responses.

RESPONSE TO REFEREE REPORT 2

The authors have satisfactorily responded to the criticism of the reviewers and revised the script according to the comments. Therefore, publication of the script in Nature Communications is recommended.

Signed by Senthil Kumar Kuppusamy

Author's response: We sincerely thank the reviewer for the positive evaluation and the recommendation for publication in *Nature Communications*. We truly appreciate the time and effort devoted to reviewing our manuscript and are grateful for the constructive feedback, which helped us improve the quality and clarity of the work.

Reviewer #3 (Remarks to the Author):

The authors have conducted a careful revision, satisfactorily addressing my previous concerns. The manuscript now meets the scientific rigor and presentation standards required for publication in *Nature Communications*.

Author's response: We sincerely thank the reviewer for the positive assessment of our revised manuscript and for recognizing the improvements made. We are grateful for the constructive feedback that helped us enhance both the scientific rigor and clarity of the presentation.

MANUSCRIPT NCOMMS-24-69634

“High sensitivity pressure and temperature quantum sensing in organic crystals”

REVIEWER’S REPORT

The manuscript by H. Singh and coauthors reports a study about temperature and pressure sensing by means of a molecular organic radical crystal (pentacene-doped terphenyl crystal). The experiments are carried out in a custom-made setup for Optically Detected Magnetic Resonance (ODMR: laser excitation, fluorescence readout and additional radiofrequency/microwave excitation, RF/MW). The data presented are very clear, accurate and, overall, of good quality. The comparison with the pressure and temperature sensing performed over different spin centers (such as – but not limited to - Nitrogen-Vacancy centers) allows to appreciate the potential of molecular spins as sensors. Notably and very interestingly, the final part of the manuscript contains DFT results which support the data and suggest the possible origin (at molecular orbital level) of the advantage of molecular spins for sensing pressure and temperature if compared to other magnetic spin impurities.

The protocol used for sensing is based on a continuous-wave optical excitation combined with RF/MW excitation, which seems to be delivered in continuous-wave as well (see my points 9 and 10 below). This means that, according to the definition reported in Ref. [1], a type I quantum sensor (quantized and individually-addressable energy levels) is realized. Moreover, it seems that another preprint by almost the same group of authors (Ref. 34) is showing quantum sensing of magnetic fields realized with a similar pentacene-doped crystal and ODMR approach but with a fully pulsed-wave mode (i.e. both optical and RF/MW pulses), i.e. as a type II sensor (always according to the definition in Ref. [1]).

In my opinion the manuscript, in its actual version, is for sure reporting a molecular-based pressure and temperature quantum sensor, highlighting potential advantages with respect to other paramagnetic spin defects or impurities and a possible explanation about the origin of such advantages. Overall, the results meet the level required for a publication in Nature Communications. However, at the same time, I have several points the authors must necessarily address before recommending the manuscript for publication.

Please find below additional points and comments:

- 1) References. In my opinion there are some missing references concerning sensing or quantum sensing of pressure and temperature. For instance, Ref. (5) in the caption of Table I, although mentioned in the table and used for comparison, is not appearing in the reference list. The same seems to occur also for Ref. (6), (8) and (9) of the same Table I. These works should be included in the reference list and properly cited (in the introduction and/or in the discussion part) since the authors are basing on their results to show the potential of their molecular sensor.

Here's possible other works the authors could consider to cite:

- <https://www.nature.com/articles/s41467-024-52272-y>
- <https://www.tandfonline.com/doi/full/10.1080/26941112.2021.1964926>

It seems also that Ref [33] of the manuscript has been recently published and it is no more a preprint:

<https://journals.aps.org/prl/abstract/10.1103/PhysRevLett.133.120801>

The introduction mentions the potential of molecular spins systems for quantum sensing. In my opinion, the authors are missing some recent developments on both the theoretical as well as the experimental side, which should be added to the references:

- <https://iopscience.iop.org/article/10.1088/2058-9565/ad985e>
- <https://www.nature.com/articles/s41534-024-00838-5>
- <https://www.sciencedirect.com/science/article/abs/pii/S0304885319311199?via%3Dihub>
- <https://www.nature.com/articles/s41565-024-01724-z>
- <https://pubs.acs.org/doi/10.1021/jacs.2c07692>
- <https://pubs.aip.org/aip/jcp/article/158/16/161103/2885321/Toward-quantum-sensing-of-chiral-induced-spin>

- 2) Doping level. The authors mention that the doping level of the crystal can be eventually high (introduction). However, the concentration used in the experiments is 0.1% corresponding to a collection of 10^9 spins and a volume of $2.6 \cdot 10^{-5} \text{mm}^3$ (Section II). This is giving FWHM linewidth of $\approx 4.8 \text{ MHz}$. This concentration value is not so small and even so large. Could the authors comment on how the linewidth is affected by the concentration and on how much the concentration could be, in principle, increased or further decreased?
- 3) Sample size. The sample shown in Fig. 1.c has a remarkably large size for a single crystal. However, the fraction of the volume used for experiments is only $2.6 \cdot 10^{-5} \text{mm}^3$ which means that most of the volume available is effectively not necessary and not used during experiments. The same holds by considering the sample volume reported in Supplementary Material (Experimental Setup section). Which are the factors limiting the use of a larger sample volume? How does the RF/MW field inhomogeneity affect the FWHM linewidth?
- 4) Cost reported for sample in Fig. 1.c. The cost for the large crystal is estimated in 2.25 US dollars accounting only for materials. I really appreciate the comparison in terms of cost of production between a synthetic crystal and an artificial diamond (such as NV centers). However, in my opinion the comparison could be further extended and detailed. Possible points of interest for considering the cost might be:
 - artificial diamonds are grown by chemical vapour deposition or by high-pressure high-temperature methods. Nitrogen and vacancies can be then created by ion implantation or neutron irradiation followed by thermal annealing. This initial difficulty can be overcome considering that there are specialized companies offering commercial NV centers.
 - molecular spins offer an easier way of fabrication due to chemical synthesis. Here, however, raw materials with adequate quality and purity might be necessary and specific chemical techniques and chemical facilities for purification or making reactions. The recipe could require several complex steps and could be not so easy to be obtained or optimized at the beginning. The yield of the reaction could be not so high.

Normalizing the cost over the mass of product might be misleading as well. An alternative could be to normalize over the density of the sample or on the spin concentration obtained (spins centers per unit of volume). This extended discussion could be added in a dedicated section of the Supplementary Material.

- 5) Fig. 2. Panel A and B show the evolution of the signal for the XY and the XZ transitions. It seems that there is a slight broadening of the signals as the temperature is increased from 100 K to 300 K. However, Fig. 1.D and 1.E suggest the broadening of the lines occurs at low temperature, not at 300 K. Can the authors comment on this point?
- 6) Section III, the change in the spectral frequency df/dT (and its equivalent for pressure sensing df/dP). It is stated that this quantity is material-specific and independent by the measurement apparatus or light collection efficiency. The temperature (or the pressure) is changed in the environment (cryostat or pressure cell) it is a global macroscopic experimental parameter of the whole sample (not a local one). Moreover, the setup it is said to be not optimized for sensitivity and that the collection efficiency affects the measured signal. In my opinion, the sample size (number of spins), the laser power, the RF/MW power can affect the absolute signal measured. So, one might expect df/dT (or df/dP) to be affected by the setup and experimental conditions as well. Could the authors better clarify this point?
- 7) End of the Section III. It is mentioned that the sensitivity $\eta = \sigma\sqrt{\tau}/\frac{dS}{dT}$ depends on many experimental parameters, including number of spins and collection efficiency so, it ultimately depends by the apparatus and experimental conditions. I agree with the definition of sensitivity used by the authors in the manuscript (which includes also the change in the spectral frequency). In my opinion introducing the noise floor level and the integration time with the mentioned definition of sensitivity can give a better comparison among different experiments and, eventually, setups and samples. This is because it can be a way to normalize the change in the spectral frequency over different experiments and apparatus. Could the authors comment on this point?
- 8) Table I. I appreciate the comparison given in Table I and its extended version in Supplementary Material. Comparing different solid-state platforms, as mentioned by the authors at the end of section II, is of course non trivial. In my opinion the reader would benefit in having more details about the available experimental parameters for each line of table I, such as: laser power, collection efficiency, number of spins of each sample. This could be part of an extended Table I in Supplementary Material.
- 9) Protocol used for quantum sensing. The manuscript is citing a preprint by almost the same group of authors (Ref. 34) in which quantum sensing of magnetic field is experimentally realized with a similar pentacene-doped crystal and using again an ODMR approach. The sensing protocol in Ref. 34 seems to be fully pulsed-wave in the sense that both optical pulses and RF/MW pulses are used, while in the present manuscript the optical excitation is given in continuous wave. This seems to hold also for the RF/MW excitation controlling the metastable spin triplet (except, maybe, for the operation of the switch of their setup, which I believe it is necessary for using the lock-in). This is further confirmed by a statement by the authors about the low optical contrast obtained and by the possibility to increase it using pulsed optical excitation.
Could the authors explain why did they choose to use a continuous wave protocol instead of a fully pulsed-wave one? Is there any advantage in running the sensor for pressure or temperature this way? In my opinion, running the sensor in pulsed-wave and exploiting its quantum coherence (as type II

sensor according to Ref. [1]) would give more convincing evidence for a quantum sensing protocol (not only a larger contrast).

- 10) Experimental setup. The experimental setup is described in detail. However, there is no information on the optical power reaching the sample or on the RF/MW ones. The reader might benefit in such additional technical information about the protocols and on how the excitations are given and which is their timing or duration (see also the introduction and point 9 above). Adding details about the role of the lock-in and on how the contrast is extracted from experimental data might help the audience which is not familiar with ODMR. This information could be added in Supplementary Material.

ADDITIONAL MINOR POINTS

- 11) It could be helpful to add a guide for eyes also in Fig. 3.A and 3.B, in a way similar to what done in Fig. 2.A and 2.B.

MANUSCRIPT NCOMMS-24-69634A

“High sensitivity pressure and temperature quantum sensing in pentacene-doped p-terphenyl single crystals”

REVIEWER'S REPORT

I have carefully checked the revised version of the manuscript by H. Singh and coauthors as well as their reply to my points. In my opinion, the authors have addressed all my previous points and to have also followed some of my suggestions.

Therefore, according to my previous report and assessment, I can recommend the paper for publication in Nature Communications in its actual revised version.

However, as a final remark and advice for the future, I strongly suggest the author to fully report the whole comments of a Reviewer when addressing them in their reply, without cutting or omitting parts of them. More specifically, the original numeration of part of my comments appeared to be wrong and it was a little hard to compare them with the authors' replies. This additional effort from my side could have been easily avoided.

There are several reasons behind this advice:

- i) The peer review file will be made available in case of publication. It would be much and much better for the readers to be able to track the whole revision history (i.e., full comments and replies). No matters about the length.
- ii) This improves the readability of the revisions made by the authors
- iii) apparently, a similar thing was not done with the comments of the other Reviewers. This could be misunderstood for a lack of consideration towards one of the Reviewers
- iv) The authors should know that omitting part of the considerations and motivations given by a Reviewer in his/her report might somehow hide or mask his/her critical thinking on the manuscript.